# ILLUSION: UNVEILING TRUTH WITH A COMPREHENSIVE MULTI-MODAL, MULTI-LINGUAL DEEPFAKE DATASET

**Kartik Thakral**[*,1]**, Rishabh Ranjan**[*,1]**, Akanksha Singh**[1,2]**, Akshat Jain**[1]**,
Mayank Vatsa**[1]**, and Richa Singh**[1]
[1]IIT Jodhpur, India, [2]IISER Bhopal, India

## ABSTRACT

The proliferation of deepfakes and AI-generated content has led to a surge in media forgeries and misinformation, necessitating robust detection systems. However, current datasets lack diversity across modalities, languages, and real-world scenarios. To address this gap, we present ILLUSION (Integration of Life-Like Unique Synthetic Identities and Objects from Neural Networks), a large-scale, multi-modal deepfake dataset comprising 1.3 million samples spanning audio-visual forgeries, 26 languages, challenging noisy environments, and various manipulation protocols. Generated using 28 state-of-the-art generative techniques, ILLUSION includes faceswaps, audio spoofing, synchronized audio-video manipulations, and synthetic media while ensuring a balanced representation of gender and skin tone for unbiased evaluation. Using Jaccard Index and UpSet plot analysis, we demonstrate ILLUSION's distinctiveness and minimal overlap with existing datasets, emphasizing its novel generative coverage. We benchmarked image, audio, video, and multi-modal detection models, revealing key challenges such as performance degradation in multilingual and multi-modal contexts, vulnerability to real-world distortions, and limited generalization to zero-day attacks. By bridging synthetic and real-world complexities, ILLUSION provides a challenging yet essential platform for advancing deepfake detection research. The dataset is publicly available at `https://www.iab-rubric.org/illusion-database`.

## 1  INTRODUCTION

The emergence of social media platforms has fundamentally transformed our mode of communication and information dissemination. Platforms such as Facebook, YouTube, Instagram, and TikTok, which boast billions of users worldwide, have expanded the scope of shared content beyond text to include images, videos, and other forms of multimedia Narayan et al. (2022a); Ranjan et al. (2024b). This shift has precipitated a surge in the volume of multi-modal content accessible online. As social networks evolve rapidly, they have emerged as the primary conduit for disseminating user-generated multi-modal content. The data circulating on these social networks is predominantly multi-modal, encompassing videos,

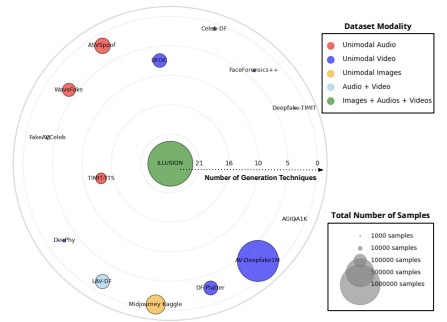

Figure 1: Comparative analysis of the proposed dataset with existing ones based on modalities, size, and manipulations.

audio, and images. With their billions-strong user base, these platforms generate enormous data every minute. Nonetheless, the rise of social media and multi-modal content has concurrently fueled an upsurge in the spread of deepfakes and synthetic media fabricated by deep learning techniques. The advancements in generative techniques like Generative Adversarial Networks (GANs), Variational Auto-Encoders (VAEs), and diffusion-based models, have significantly enhanced the realism of

---

[*]Equal Contributions.

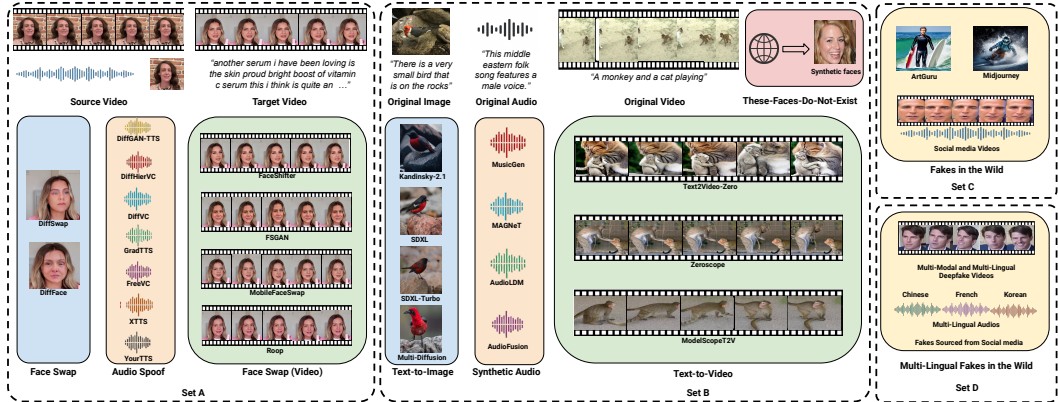

Figure 2: Visual representation and organization of each subset in the proposed ILLUSION dataset.

synthetically generated content, making it more convincing to the untrained eye Nirkin et al. (2019); Thakral et al. (2024); Korshunova et al. (2017); Rombach et al. (2022); Ranjan et al. (2023b). These AI foundational models and diffusion-based Generative AI (GenAI) models have exhibited unparalleled competence in comprehending and generating human-like videos, images, and sounds (Ranjan et al., 2024a). In 2023, 95,820 deepfake videos were circulated online, which is a 550% increase from 2019 (Deepfake Statistics). Projections indicate that by 2026, up to 90% of online content may be synthetically generated (Synthetic AI Stats). This rapid surge highlights the urgent need for robust deepfake detection techniques. In response, numerous researchers have developed methods specifically targeting identity-based deepfakes (Afchar et al., 2018; Nguyen et al., 2019; Agarwal et al., 2019; Khalid & Woo, 2020; Thakral et al., 2023; Chhabra et al., 2024; Ranjan et al., 2023a).

The success of deepfake detection depends on the quality of the dataset, including diversity in modalities, generation methods, and the realism of samples. Table 1 offers a comprehensive overview of public deepfake detection datasets. Early datasets such as DF-TIMIT (Korshunov, 2018), Face-Forensics++ (Rössler et al., 2018), Celeb-DF (Li et al., 2020), WildDeepfake (Zi et al., 2020), and DeeperForensics-1.0 (Jiang et al., 2020) were relatively small and unimodal, while larger datasets like FFIW$_{10k}$ (Zhou et al., 2021), KoDF (Kwon et al., 2021), and DF-Platter (Narayan et al., 2023) primarily focused on visual manipulations. Unimodal audio-based forgeries were introduced in ASVSpoof (Wang et al., 2020b), WaveFake (Frank & Schönherr, 2021), and TIMIT-TTS (Salvi et al., 2023), whereas multi-modal deepfake detection emerged in DFDC (Dolhansky et al., 2020) and FakeAVCeleb (Khalid et al., 2021). More recently, large-scale multi-modal datasets such as LAV-DF (Cai et al., 2022) and AV-Deepfake1M (Cai et al., 2024) introduced audio-video manipulations, though they lacked non-identity-based AIGC.

Existing data sets have mainly focused on a few specific generation techniques. To bridge this gap, we introduce *ILLUSION*: **I**ntegration of **L**ife-**L**ike **U**nique **S**ynthetic **I**dentities and **O**bjects from **N**eural Networks (ILLUSION Leaderboard), a novel multi-modal, multi-lingual deepfake dataset divided across four sets (visualized in Figure 2). Set A is an identity forgery dataset with audio-video synchronized. Set B incorporates AI-generated synthetic data covering three media modalities: image, audio, and video. Set C, a test set, includes a pool of real-world AI-generated content (AIGC) sampled from different sources and set D includes multi-lingual and multi-modal deepfake samples spanning over 26 different languages. The dataset is prepared with continuous usage of 40 GPUs, accounting for 2000 GB of cumulative memory. With over 800 GBs in size, the dataset contains over 1.3 million samples encompassing the four sets. To the best of our knowledge, this is one of the largest datasets containing vast variability of generation methods, different modalities, multiple languages, and various challenges (refer Figure 1).

The proposed comprehensive dataset provides diverse AI-generated content to serve as a valuable asset for research in detecting AI-generated media varying in input modality, generation models, different languages, and content type. To assess and analyze the utility of our dataset, we conduct extensive experiments and benchmark using 11 baseline deepfake methods and analyze their performance when tested in different settings. This dataset is presented as a leaderboard (ILLUSION Leaderboard) for

Table 1: Details of publicly available deepfake datasets.

| Dataset Name | Year | Real Samples | Fake Samples | | | Total Samples | Generation Techniques | Identity Swapping | AI Generated Content | Multi-Lingual | AI-Swap-Lingual |
|---|---|---|---|---|---|---|---|---|---|---|---|
| | | | Images | Audio | Video | | | | | | |
| Deepfake-TIMIT (Korshunov, 2018) | 2018 | 640 | N/A | N/A | 320 | 960 | 2 | ✓ | × | × | × |
| FaceForensics++ (Rossler et al., 2019) | 2019 | 1000 | N/A | N/A | 4,000 | 5000 | 4 | ✓ | × | × | × |
| Celeb-DF (Li et al., 2020) | 2020 | 590 | N/A | N/A | 5,639 | 6,229 | 1 | ✓ | × | × | × |
| DFDC (Dolhansky et al., 2020) | 2020 | 23,654 | N/A | N/A | 104,500 | 128,154 | 8 | ✓ | × | × | × |
| DeeperForensics-1.0 (Jiang et al., 2020) | 2020 | 50,000 | N/A | N/A | 10,000 | 60,000 | 1 | ✓ | × | × | × |
| ASVSpoof (Yamagishi, 2019) | 2021 | 16,492 | N/A | 148,148 | N/A | 164,640 | 19 | ✓ | × | × | × |
| WaveFake (Frank & Schönherr, 2021) | 2021 | 0 | N/A | 117,985 | N/A | 117,985 | 6 | ✓ | × | × | × |
| FakeAVCeleb (Khalid et al., 2021) | 2021 | 500 | N/A | 500 | 9000 | 10,000 | 4 | ✓ | × | × | × |
| TIMIT-TTS (Salvi et al., 2023) | 2022 | 430 | N/A | 80,000 | N/A | 80,430 | 14 | ✓ | × | × | × |
| DeePhy (Narayan et al., 2022b) | 2022 | 100 | N/A | N/A | 5,040 | 5,140 | 3 | ✓ | × | × | × |
| LAV-DF (Cai et al., 2022) | 2022 | 36,431 | N/A | 33,176 | 65,997 | 136,304 | 2 | ✓ | × | × | × |
| Midjourney Kaggle (Iulia Turc, 2022) | 2022 | 0 | 250,000 | N/A | N/A | 250,000 | 1 | × | ✓ | × | × |
| DF-Platter (Narayan et al., 2023) | 2023 | 764 | N/A | N/A | 132,496 | 133,260 | 3 | ✓ | × | × | × |
| AV-Deepfake1M (Cai et al., 2024) | 2023 | 286,721 | N/A | N/A | 860,039 | 1,146,760 | 3 | ✓ | × | × | × |
| AGIQA1K (Zhang et al., 2023) | 2023 | 0 | 1,080 | N/A | N/A | 1,080 | 2 | × | ✓ | × | × |
| TWIGMA (Chen & Zou, 2023) | 2024 | 0 | 800,000 | N/A | N/A | 800,000 | N/A | × | ✓ | × | × |
| DeePhyV2 (Thakral et al., 2025) | 2025 | 100 | N/A | N/A | 8,960 | 9,060 | 4 | ✓ | × | × | × |
| **ILLUSION (Proposed)** | **2025** | **139,740** | **905,548** | **27,244** | **299,454** | **1,371,986** | **28** | ✓ | ✓ | ✓ | ✓ |

researchers to test the performance of their algorithms and foster the development of generalizable detection methods. The primary contributions of our work are summarized below:

- We introduce a multi-modal deepfake dataset developed using 28 GenAI models grounded in GANs, VAEs, Transformers, and Diffusion-based models, spanning image, audio, and video modalities. This dataset is partitioned into four distinct sets.

- The dataset encompasses identity manipulations, where forgery can manifest across audio, video, or both. This set is seamlessly synchronized across audio-visual channels and maintains a balance in terms of sex and skin tone.

- The dataset also includes AI-generated content (AIGC) produced by various text-to-modality models across image, audio, and video domains. It encompasses a subset of entirely synthetic faces. Additionally, the dataset features real-world deepfakes, designed to evaluate detection algorithms in a practical context that spans multiple modalities and languages.

- We benchmark the proposed dataset using state-of-the-art detection methods and conduct a comprehensive analysis of their performance across a range of challenging protocols.

## 2 THE ILLUSION DATASET

In this paper, we introduce *ILLUSION:* **I**ntegration of **L**ife **L**ike **U**nique **S**ynthetic **I**dentities and **O**bjects from **N**eural networks, a comprehensive large-scale multi-modal deepfake dataset[1]. The dataset comprises 1,376,371 samples across image, audio, video, and synchronized audio-video modalities, making it the largest multi-modal resource available in the deepfake literature. The dataset is organized into four subsets: Set A, Set B, Set C, and Set D. Set A includes identity manipulations featuring faceswaps, voice spoofs, and both. Set B comprises synthetically generated media, including images and videos of sceneries, objects, situations, and music audio. This set also incorporates synthetic faces generated from the website (TPDNE). Set C encompasses real-world testing samples, i.e., **Fakes in the Wild**, generated using proprietary generative models, and Set D is a multi-lingual multi-modal deepfake testing set. ILLUSION is produced using 28 distinct generative models, encompassing both open-source and proprietary systems. Unlike many publicly available deepfake datasets that suffer from imbalances in sex and skin tone (Nadimpalli & Rattani, 2022; Xu et al., 2022a), the ILLUSION dataset ensures equitable representation across these subgroups. An overview of the dataset is provided in Figure 2.

### 2.1 DATASET STATISTICS AND ORGANIZATION

This section discusses the statistics and organization of each set of the proposed ILLUSION dataset. Table 2 presents the set-wise statistics.

**Set A:** This set comprises a total of 13 identity manipulations, generated from 200 unique identities sampled from the CelebV-Text dataset (Yu et al., 2023). Each audio, video, and audio-synchronized

---

[1]The collection and generation of the dataset have been approved by the Institutional Ethics Review Committee. Access will be limited to academic institutions for research purposes.

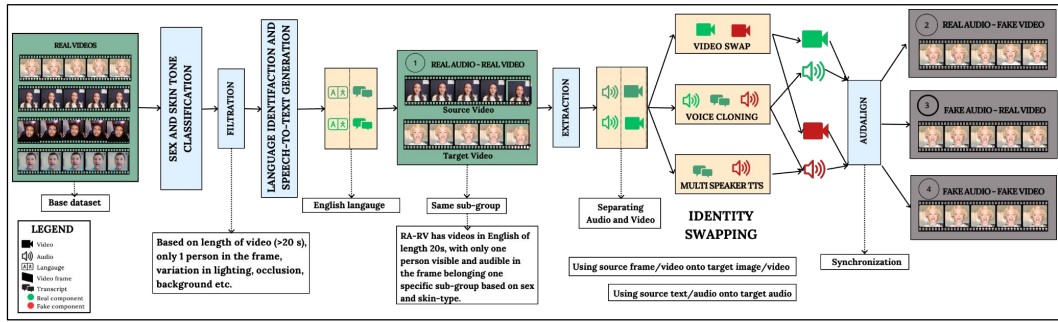

Figure 3: Illustration of generation pipeline employed in set A for identity swaps.

clip in this set is 20 seconds long. The samples in this set incorporate variations such as occlusions (e.g., hats, glasses, beards, etc.), body movements, and lighting conditions, thereby ensuring a diversity of variations in the dataset. This set is categorized into four classes: (i) Real Audio- Real Video (RA-RV), (ii) Real Audio and Fake Video (RA-FV), (iii) Fake Audio and Real Video (FA-RV), and (iv) Fake Audio and Fake Video (FA-FV) (as visualized in Figure 3). Table 2 summarizes the number of samples available in each class of Set A. Manipulations in images and videos for each of these classes are introduced by swapping the faces of the source identity onto the target video. For this purpose, we employ 6 different face-swapping models, namely, MobileFaceSwap (Xu et al., 2022b), FSGAN (Nirkin et al., 2019), FaceShifters (Li et al., 2019b), ROOP (Roop, 2023), DiffFace (Kim et al., 2022), and DiffSwap (Zhao et al., 2023). The audio deepfakes are created using the English transcription generated through the MMS model (Pratap et al., 2024). This transcription is then employed by Text-to-Speech systems to create identity-swapped voice clones. We utilize 7 different audio-generative models, namely, FreeVC (Li et al., 2023), XTTS (Eren & The Coqui TTS Team, 2021), DiffVC (Popov et al., 2021b), DiffHierVC (Choi et al., 2023), YourTTS (Casanova et al., 2022), DiffGAN-TTS (Liu et al., 2022), and GradTTS (Popov et al., 2021a) for voice-swapping. For classes RA-FV, FA-RV, and FA-FV, we employ Alignment, a fingerprinting-based model to ensure seamless synchronization between the audio and video, thereby enhancing the realism of the generated fake. The details of all the models utilized in this set are available in the Appendix.

**Set B:** This set comprises a total of 523,222 entirely synthetic samples and their 138,708 real counterparts, generated through 11 open-source models and one closed-source model. This set includes images, audio, and videos primarily generated using diffusion models and transformers. For the generation of synthetic images, we employ the images and their corresponding prompts from the training set of the COCO dataset (Lin et al., 2014) to generate using four text-to-image generative models. These models include Stable Diffusion-XL (Podell et al., 2023), Kandinsky 2.1 (Razzhigaev et al., 2023), MultiDiffusion (Bar-Tal et al., 2023), and SDXL-Turbo (Sauer et al., 2023). We also collected

Table 2: **Dataset Statistics:** In Set A, audio and images are extracted from videos for manipulation, with three compression levels (raw, C23, and C40).

| Sets | Modality | Generation Methods | Real Samples | Fake Samples | Total |
|------|----------|--------------------|--------------|--------------|-------|
| Set A | Audio | 8 | N/A | 6,400 | 6,400 |
| | Images | 2 | N/A | 403,200 | 403,200 |
| | Videos | 4 | 1,032 | 278,400 | 279,432 |
| Set B | Audio | 4 | 5,211 | 20,844 | 26,055 |
| | Images | 4 | 118,287 | 473,148 | 591,435 |
| | Videos | 3 | 7,010 | 21,030 | 28,040 |
| | Faces | 1 | 8,200 | 8,200 | 16,400 |
| Set C | Images | 2 | N/A | 21,000 | 21,000 |
| | Videos | N/A | N/A | 24 | 24 |
| Set D | Audios | N/A | 1600 | 2560 | 4160 |
| | Videos | N/A | 100 | 125 | 225 |
| Total | | 28 | 141,440 | 1,234,931 | 1,376,371 |

8,200 synthetic face images from "This Person Does Not Exist", ensuring a balance in terms of sex and skin tone. These identities are entirely synthetic, have not been swapped, and do not exist in the real world. To generate synthetic audio, we utilize audios and corresponding captions from the MusicCaps dataset (Agostinelli et al., 2023) and generate 5,211 synthetic audio samples each from three text-to-audio generative models and one audio-to-audio model, namely, AudioLDM (Liu et al., 2023), MusicGen (Copet et al., 2023), MAGNeT (Ziv et al., 2024), and Audio Diffusion . Similarly, we also generated synthetic videos using three text-to-video generative models, namely, Text2Video-Zero (Khachatryan et al., 2023), ModelScopeT2V (Wang et al., 2023a), and Zeroscope. For this, we borrow the corresponding caption for each video from the training set of MSRVTT dataset (Xu et al., 2016) and utilize it to generate 7,010 synthetic videos. The details of all the models utilized in this set are available in the Appendix.

**Set C:** This set serves as a real-world test set, comprising 21,024 fake samples. It is a curated collection of viral deepfake videos circulated on social media platforms and samples generated using

| Dataset | AGIQA-1K | AV-Deepfake1M | DF-Platter | Midjourney-Kaggle | LAV-DF | DeePhy | TIMIT-TTS | FakeAVCeleb | ForgeryNet | KoDF | DiffusionFace | DIRE | WaveFake |
|---|---|---|---|---|---|---|---|---|---|---|---|---|---|
| Jaccard Index | 0.00 | 0.03 | 0.07 | 0.04 | 0.00 | 0.07 | 0.00 | 0.03 | 0.05 | 0.03 | 0.03 | 0.00 | 0.00 |

Table 3: Jaccard Index Comparison of ILLUSION with existing deepfake datasets that highlights the uniqueness of the proposed dataset.

proprietary models such as Midjourney and ArtGuru. The former includes 24 identity-swapped videos. The latter consists of synthetic images generated through a premium API and a web interface, respectively. For Midjourney, we utilize prompts from the COCO dataset validation set, and for each prompt, we obtain four corresponding images, resulting in a total of 20,000 images. ArtGuru, specializes in generating identity-driven images for a given prompt. Therefore, we randomly select 1000 prompts from class "person" from COCO dataset to generate a total of 1000 images.

**Set D:** This is a multi-lingual, multi-modal real-world test set containing 4,385 samples curated from social media across 26 languages including French, German, Italian, Chinese, Korean, Arabic, Japanese, Tamil, Kannada, Oriya, Hindi, Sanskrit, Latin, Punjabi, and Gujarati. It is divided into D.1 with 4,160 web-curated samples and D.2, a subset of 225 deepfake samples annotated with four classes (RA-RV, RA-FV, FA-RV, FA-FV), with synchronized audio and video from diverse origins.

## 2.2 SIZE AND FORMAT

The ILLUSION dataset is ∼800 GB, with Set A clips lasting 20 seconds. All facial images are either synthetically generated or sourced from publicly available datasets with appropriate licensing. Videos are stored in MPEG4.0 format at a resolution of 512×512, preserving the original frame rate. Consistency is maintained across the dataset through uniform resolution, compression, and generation techniques, with H.264 compression applied at c23 and c40 levels. Skin tones are categorized into four Fitzpatrick scale bins (see Appendix) and combined with two sexes to form eight sub-groups—identity swaps occur only within these groups. In Set B, videos are generated at 24 frames per second using Text2Video-Zero and ZeroScope, and at 40 frames per second using ModelScopeT2V. Set C, curated from various sources, varies in resolution and length; however, images from MidJourney and ArtGuru consistently maintain resolutions of 1024×1024 and 512×512, respectively.

## 2.3 AUDIO AND VISUAL QUALITY ASSESSMENT AND COMPARISON

To assess the visual quality of ILLUSION, we compute the BRISQUE score (Mittal et al., 2012) across all four sets (Figure 4), yielding a mean score of 38.04 on a 0 (best) - 100 (worst) scale. Table 4 compares BRISQUE scores with Face-Forensics++, CelebDF, DFDC, OpenForensics, and DF-Platter, approximated from (Narayan et al., 2023). Additionally, we report FAD scores (Kilgour et al., 2018), capturing audio quality, with a mean FAD of 9.43. These scores highlight that the ILLUSION is of high quality and is challenging with multiple covariates. Additionally, we

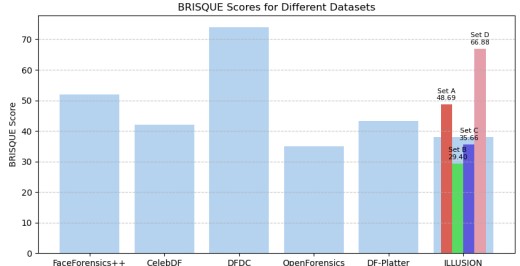

Figure 4: Comparing Brisque Score of ILLUSION with other datasets.

compute the Jaccard Similarity Index to quantify the overlap between ILLUSION and other datasets. As shown in Table 3, the results indicate minimal overlap, with values as low as 0.03 (FakeAVCeleb, KoDF) and 0.07 (DF-Platter). Notably, AGIQA-1K, LAV-DF, and DIRE show no overlap (0.00), highlighting ILLUSION's unique generative diversity.

## 2.4 COMPUTATIONAL SETUP

In Set A, we utilize a total of 13 generation methods to produce identity-swaps across image, audio, video, and audio-video synchronized modalities. This process is facilitated by Nvidia A100 with 16 GPUs, each with 80GBs of memory. Set B is generated through 11 open-

Table 4: Audio and visual quality assessment of ILLUSION dataset.

| Dataset Split | Vision (Mean Brisque Score) | | | Audio (Mean Fréchet Audio Distance) | | |
|---|---|---|---|---|---|---|
| | Train Set | Test Set | Overall | Train Set | Test Set | Overall |
| Set A | 49.08 | 42.98 | 48.69 | 6.37 | 7.30 | 6.41 |
| Set B | 29.43 | 29.29 | 29.40 | 6.64 | 6.58 | 6.55 |
| Set C | N/A | 35.66 | 35.66 | N/A | N/A | N/A |
| Set D.1 | N/A | N/A | N/A | N/A | 40.39 | 40.39 |
| Set D.2 | N/A | 66.88 | 66.88 | N/A | N/A | N/A |
| | Overall: 38.04 | | | Overall: 9.43 | | |

source and one closed-source generative models, utilizing two Nvidia A40 GPUs, each with 48GBs

of memory, and three Nvidia DGX stations, each equipped with four V100 GPUs of 32GB memory. Set C comprises samples generated from two proprietary models, produced on 2 Nvidia 3090 GPUs, each with 24 GBs of memory. The benchmarking experiments for the dataset are conducted on 2 A40, each with 48GBs of memory, and 6 A30 GPUs, each with 24GBs of memory, in a multi-GPU setup.

# 3   EXPERIMENTAL SETUP

This section outlines the training and testing protocol established for the proposed ILLUSION dataset, followed by a discussion on the deepfake detection methods and evaluation metrics employed for benchmarking. The proposed dataset is designed to address the following pivotal research questions:

**RQ1:** How effective are the detection systems in detecting multi-modal identity-swaps?

**RQ2:** How effective are the detection systems in identifying synthetically generated media?

**RQ3:** How robust and reliable are the current state-of-the-art detection algorithms when deployed in real-world scenarios?

**RQ4:** Is it feasible to detect identity swaps and synthetic media in a zero-day attack setting?

**RQ5:** Is it possible to successfully trace back the source of a given deepfake?

## 3.1   EVALUATION PROTOCOLS

The ILLUSION dataset is composed of four sets. Sets A and B are partitioned into training and testing subsets in a ratio of 3:1. The training data is split into a 9:1 ratio to divide into train and validation data. To mitigate the skew between the "Real" and "Fake" classes in set A, we borrow an additional 144 videos (18 subjects/sub-group) from the CelebV-Text dataset. In contrast, set C and D is exclusively a test set. For all the videos in Set A, we extract 10 frames from each fake video and all from each real video. For Set B, we pick 24 frames each from the generative models and select every sixth frame from real videos. Further, for synthetic images generated from four text-to-image models, we repeat their corresponding real images four times. This approach addresses the imbalance between the dataset's real and fake samples.

**Protocol 1 - Multi-modal Deepfake Detection:** This protocol utilizes Sets A and B, each with their respective training and test sets. Set A is divided in a subject-disjoint manner, incorporating 160 subjects in the training set (20 subjects/sub-group) and 40 subjects in the test set (5 subjects/sub-group). The audio, video, and multi-modal deepfake detection models are then trained and tested on the samples from Set A. The results are presented in three compression settings - raw, C23, and C40 - to facilitate the assessment of deepfake quality in the dataset relative to existing datasets (results are shown in the Appendix). For Set B, images borrowed from the COCO dataset (Lin et al., 2014) and BFW dataset (Robinson, 2022) serve as real samples corresponding to the fake samples generated using text-to-image models and synthetic faces, respectively. Similarly, audio and video samples from the MusicCaps dataset (Agostinelli et al., 2023) and the MSRVTT dataset (Xu et al., 2016) are used as real samples corresponding to fake samples generated using text-to-audio and text-to-video models. We extract 24 frames from fake videos to classify synthetic videos and select every 6th real video frame to maintain data balance.

**Protocol 2 - Zero-shot/Zero-day Generalization:** The primary aim of this protocol is to test the generalizability of detections on new or unseen generation methods. The detection models are initially trained on the train set of Set A and subsequently tested on the test set of Set B. The performance of the models is also evaluated in a vice-versa setting.

**Protocol 3 - Generalization on Real-World Deepfake Media:** This protocol assesses the performance of existing state-of-the-art models on real-world deepfake samples. Here, the models are trained on the train set of Set A, Set B, or both, and their performance is evaluated on Set C.

**Protocol 4 - Performance on Model Attribution:** The final protocol presents a challenging model attribution task, i.e. to predict the generative technique used to create the input deepfake. The models are trained and tested on fake samples generated from each technique from Set A and Set B. The detection models are evaluated separately for image, video, and audio modalities.

Table 5: Classification performance for visual components of the dataset obtained by varying the training and testing sets.

| Trained On | Models | Set A | | | | Set B | | | | Set C (All Fake) |
|---|---|---|---|---|---|---|---|---|---|---|
| | | Accuracy | Accuracy-Fake | Accuracy-Real | AUC | Accuracy | Accuracy-Fake | Accuracy-Real | AUC | Accuracy |
| Set A | F3Net | 0.851 | 0.751 | 0.951 | 0.945 | 0.468 | 0.475 | 0.460 | 0.462 | 0.341 |
| | DSP-FWA | 0.984 | 0.982 | 0.986 | 0.997 | 0.428 | 0.437 | 0.418 | 0.396 | 0.214 |
| | MesoInceptionNet | 0.505 | 0.999 | 0.882 | 0.883 | 0.505 | 0.988 | 0.473 | 0.487 | 0.991 |
| | Xception | 0.851 | 0.846 | 0.856 | 0.920 | 0.497 | 0.749 | 0.283 | 0.515 | 0.633 |
| Set B | F3Net | 0.498 | 0.008 | 0.993 | 0.497 | 0.981 | 0.991 | 0.970 | 0.998 | 0.717 |
| | DSP-FWA | 0.498 | 0.001 | 0.999 | 0.501 | 0.993 | 0.995 | 0.990 | 0.999 | 0.727 |
| | MesoInceptionNet | 0.495 | 0.009 | 0.987 | 0.386 | 0.757 | 0.553 | 0.967 | 0.919 | 0.045 |
| | Xception | 0.502 | 0.018 | 0.992 | 0.526 | 0.977 | 0.987 | 0.966 | 0.997 | 0.725 |
| Set A + Set B | F3Net | 0.881 | 0.926 | 0.836 | 0.958 | 0.956 | 0.986 | 0.925 | 0.994 | 0.703 |
| | DSP-FWA | 0.972 | 0.975 | 0.970 | 0.995 | 0.992 | 0.995 | 0.989 | 1.000 | 0.863 |
| | MesoInceptionNet | 0.481 | 0.013 | 0.425 | 0.701 | 0.834 | 0.806 | 0.912 | 0.948 | 0.241 |
| | Xception | 0.881 | 0.926 | 0.835 | 0.954 | 0.972 | 0.982 | 0.963 | 0.997 | 0.650 |

## 3.2 BENCHMARKING DETAILS

**DeepFake Detection Methods** We utilize four state-of-the-art video and four audio deepfake detection models to benchmark all three sets of the proposed dataset. For video deepfake detection, we employ MesoInceptionNet (Afchar et al., 2018), XceptionNet (Chollet, 2017), DSP-FWA (Li & Lyu, 2018), and F3Net (Wei et al., 2020). For audio deepfake detection, we use RawGAT-ST (Tak et al., 2021), AASIST (Jung et al., 2022), SSLModel (Tak et al., 2022), and Conformer (Gulati et al., 2020). We also benchmark the proposed dataset using multi-modal deepfake detection algorithms. Specifically, we employ methods such as MRDF (Zou et al., 2024) and FACTOR (Reiss et al., 2023). Additionally, we use an ensemble of F3Net and SSLModel, which are baseline unimodal models (referred to as unimodal ensembling), and report class-wise video-level accuracy. Benchmarking the proposed ILLUSION dataset with 11 baseline algorithms provides a comprehensive evaluation that encompasses both typical methods and type-complete approaches. The unimodal baselines focus on modality-specific behaviors, enabling a deeper understanding of how state-of-the-art algorithms perform within their respective domains (e.g., image, audio, or video). In contrast, multimodal baselines evaluate type-complete methods, capturing the interplay between multiple modalities and offering insights into cross-modal generalization and robustness. This dual benchmarking strategy ensures a balanced assessment of both specialized and holistic detection capabilities. Detailed descriptions of all these algorithms are provided in the Appendix.

**Evaluation Metrics** For models trained on image and video data, we provide frame-level accuracy and Area Under the Curve (AUC) scores. Each frame in a video classified as fake or real. We also present class-wise accuracy for additional analysis. For audio data, we report the Equal Error Rate (EER) and AUC score. For models trained on multi-modal data, such as combined video and audio, we provide video-level accuracy, using a threshold set at 50% of frames to classify a video as fake.

**Implementation Details** The DSFD detector (Li et al., 2019a) is used to extract faces from the frames of videos containing faces. For all protocols, the models are trained for 30 epochs with early stopping, and the models with the best validation accuracy are selected. We use the Adam optimizer with an initial learning rate of 0.0001. A batch size of 256 is used for distributed training.

# 4 RESULTS AND DISCUSSION

This section discusses the benchmark results obtained using the state-of-the-art deepfake detection models mentioned in Section 3.2 when trained and evaluated on the proposed ILLUSION dataset. The performance analyzed is in accordance with the protocols described in section 3.1.

**Protocol 1 - Multi-Modal Deepfake Detection:** To analyze the performance of audio and visual detection models, we trained and tested them on both Set A and Set B of the proposed ILLUSION dataset. From Tables 5 and 6, for set A, we observe that all architectures perform well for visual as well as audio detection models, with DSP-FWA achieving the best performance for visual data (99.3% accuracy on Set B) and SSLModel excelling in audio data with an EER of 0.006. A similar trend is visible in set B, where we observe that all the detection models, when trained on synthetic data, are able to achieve a promising detection performance.

We observe significant variability among the models when evaluating performance on Set C—which consists entirely of curated real-world and compressed deepfakes. For instance, DSP-FWA achieves

Table 6: Classification performance for audio components of the dataset obtained by varying the training and testing sets.

| Trained On | Models | Set A | | | | Set B | | | |
|---|---|---|---|---|---|---|---|---|---|
| | | EER | Accuracy-Fake | Accuracy-Real | AUC | EER | Accuracy-Fake | Accuracy-Real | AUC |
| Set A | RawGAT-ST | 0.443 | 1.000 | 0.000 | 0.576 | 0.470 | 0.999 | 0.004 | 0.539 |
| | AASIST | 0.049 | 0.879 | 1.000 | 0.991 | 0.430 | 0.240 | 0.922 | 0.563 |
| | Conformer | 0.025 | 0.996 | 0.975 | 0.991 | 0.562 | 0.715 | 0.081 | 0.395 |
| | SSLModel | 0.006 | 0.980 | 1.000 | 1.000 | 0.583 | 0.676 | 0.104 | 0.356 |
| Set B | RawGAT-ST | 0.381 | 0.125 | 1.000 | 0.699 | 0.032 | 0.988 | 0.939 | 0.995 |
| | AASIST | 0.261 | 0.071 | 1.000 | 0.805 | 0.020 | 0.988 | 0.970 | 0.997 |
| | Conformer | 0.821 | 0.008 | 0.825 | 0.119 | 0.005 | 0.992 | 0.994 | 0.999 |
| | SSLModel | 0.694 | 0.005 | 0.925 | 0.252 | 0.006 | 0.993 | 0.993 | 0.999 |
| Set A + Set B | RawGAT-ST | 0.025 | 0.943 | 0.975 | 0.994 | 0.035 | 0.973 | 0.958 | 0.995 |
| | AASIST | 0.050 | 0.946 | 0.950 | 0.992 | 0.027 | 0.983 | 0.951 | 0.996 |
| | Conformer | 0.030 | 0.959 | 0.975 | 0.995 | 0.020 | 0.990 | 0.959 | 0.997 |
| | SSLModel | 0.069 | 0.938 | 0.925 | 0.988 | 0.022 | 0.992 | 0.952 | 0.998 |

only 21.4% accuracy when trained on Set A (synthetic data) and tested on Set C, compared to 72.7% accuracy when trained on Set B (real-world data) and tested on Set C. This discrepancy highlights the challenges of generalizing from synthetic training data to real-world deepfake scenarios. Conversely, MesoInceptionNet exhibits unusually high accuracy on Set C due to its tendency to classify most inputs as fake. Since Set C contains only fake samples, this bias results in inflated performance metrics for MesoInceptionNet. Set C was intentionally designed as a challenging test set that closely mimics real-world deepfakes crafted for mass misinformation campaigns. Its inclusion emphasizes the necessity for detection models to handle a diverse array of generative techniques and the complexities inherent in real-world scenarios. This highlights the limitations of current approaches and showcases the forward-looking design of the ILLUSION dataset in advancing deepfake detection research.

We evaluated the effectiveness of multi-modal deepfake detection methods on Set A of the proposed ILLUSION dataset. These methods were trained on audio-video synchronized samples from Set A. The performance achieved are detailed in Table 7. Our findings show that MRDF outperforms FACTOR across all classes, notably achieving an 87.1% class-wise accuracy for the FakeAudio-FakeVideo category. Conversely, FACTOR

Table 7: Classification performance of multi-modal deepfake detection methods on set A.

| Models | RA-RV | RA-FV | FA-RV | FA-FV |
|---|---|---|---|---|
| MRDF | 0.775 | 0.446 | 0.827 | 0.871 |
| FACTOR | 0.157 | 0.352 | 0.369 | 0.413 |
| Unimodal Ensembling | 0.208 | 0.887 | 0.359 | 0.779 |

consistently underperforms, with a notable low of 15.7% accuracy for the RealAudio-RealVideo class. Unimodal Ensembling shows potential, achieving a 77.9% accuracy on the FakeAudio-FakeVideo class, but falls short for the RealAudio-RealVideo class.

To assess the impact of noise and neural compression, we evaluate baseline models trained on Sets A and B, testing them on the corrupted version of Set C (as shown in Table 1 of the Appendix). Here we observe a significant performance degradation. Additionally, we investigate different compression levels for deepfake detection models, individually training and testing them on raw, C23, and C40 compressions of Set A. From Table 2 (Appendix), DSP-FWA consistently performs well across most combinations. While there's a drop in performance when models are trained on higher quality and tested on compressed samples, those trained on C23 and C40 exhibit better generalization for both raw and C40 samples.

**Protocol 2 - Zero-Day Attack Generalization:** We evaluate the deepfake detection models on the challenging setting of zero-day attack detection. In this, each model is trained on training data from one set of the ILLUSION dataset and is tested on the test data of the other set. For evaluation on unseen attack setting, we train each model on train data of set A and test its performance on test data of set B, and vice-versa. The performance achieved is reported in Table 5 and Table 6. We clearly observe that all the visual and audio detection models, when trained on set A data and tested on set B data, consistently achieve random performance. The same observation is made for both visual and audio detection models when training data of set B is trained and evaluated on test data of set A. From this, we infer that the artifacts introduced in identity-swaps deepfakes and completely synthetic deepfakes are completely different. Due to this, the detection models trained on one is unable to generalize on the other. With this, we note that the proposed ILLUSION dataset will provide the researchers with a diverse range of deepfakes to capture variety of artifacts in training for better generalizability in real-world deployment.

**Protocol 3 - Generalization on Real-World Deepfake Media:** Since set C contains only visual deepfake media, we train the visual deepfake detection models in three different settings and report the accuracy on set C in Table 5. First, the models are trained on the image and video data of set A and then tested on set C. Then, the models are trained on set B and tested on set C. Finally, we train the models on a combination of visual data from sets A and B and test it on set C. We observe that models perform better on set C when trained on set B than when trained on set A. This behavior is observed because most samples are synthetically generated using text-to-image models like MidJourney and ArtGuru. Whereas identity swaps are very few. We also observe a slight increase in performance on set C when the detection models are trained on a combination of set A and set B.

We further perform a comprehensive evaluation of audio and multi-modal detection models on Set D, with detailed results presented in Tables 8 and 9, respecctively. The samples in Set D were assessed using audio detection models for both subsets D.1 and D.2. For Set D.1, we employed audio detection models pretrained on a combination of Set A and Set B from the ILLUSION dataset. As shown in Table 8, all architectures encountered significant challenges in generalization, with the conformer model achieving the highest AUC of 0.488. For Set D.2, we utilized MRDF, FACTOR, and Unimodal Ensembling for multi-modal baselining and report the performance in Table 9. It is evident that model performance drastically declines when evaluated on Set D, which involves multi-modal real-world fakes. Nonetheless, FACTOR outperformed Unimodal Ensembling.

Table 8: Classification accuracy of audio deepfake detection models (trained on Sets A and B) tested on Set D.1. Set D comprises audio samples from more than 26 languages.

| Models | EER | AUC |
|---|---|---|
| SSLModel | 0.578 | 0.397 |
| Conformer | 0.523 | 0.488 |
| AASIST | 0.506 | 0.471 |
| RawGAT_ST | 0.571 | 0.402 |

These results highlight the formidable challenge posed by the multi-lingual and multi-modal nature of Set D, revealing that current state-of-the-art algorithms struggle to generalize to real-world deepfakes. Our observations indicate that state-of-the-art detection models face significant difficulties when confronted with the complexity of multi-modal and multi-lingual deepfakes prevalent on social media platforms. These findings highlight the challenges and limitations these models encounter in adapting to the diverse and evolving nature of deepfake content.

Table 9: Classification performance of multi-modal deepfake detection methods on set D.2 of the ILLUSION dataset.

| Models | RA-RV | RA-FV | FA-RV | FA-FV |
|---|---|---|---|---|
| MRDF | 0.31 | 0.10 | 0.15 | 0.09 |
| FACTOR | 0.35 | 0.12 | 0.20 | 0.13 |
| Unimodal Ensembling | 0.25 | 0.09 | 0.11 | 0.05 |

**Protocol 4 - Performance on Model Attribution:** Different generation techniques introduce distinct artifacts in deepfakes (Wang et al., 2020a; Frank et al., 2020; Wang et al., 2023b). From Protocol 2, we observe that identity-swap deepfakes exhibit artifacts different from those in fully synthetic media. To analyze detection models' ability to identify the source generation technique, we report their performance in Table 10. All visual models, except MesoInceptionNet, accurately classify identity-swap deepfakes, with DSP-FWA achieving near-perfect accuracy. A similar trend is seen in audio models, where Conformer performs best.

Table 10: Model attribution on Set A

| Attribute | Models | Accuracy | AUC |
|---|---|---|---|
| Video | F3Net | 0.923 | 0.933 |
| | DSP-FWA | 0.987 | 1.000 |
| | MesoInceptionNet | 0.444 | 0.620 |
| | Xception | 0.880 | 0.832 |
| Audio | RawGAT-ST | 0.941 | 0.995 |
| | AASIST | 0.957 | 0.998 |
| | Conformer | 0.967 | 0.999 |
| | SSLModel | 0.959 | 0.998 |

For the model attribution in set B, detection models are evaluated separately for each modality. We report the performance in Table 11. For attribution in text-to-image (including synthetic faces samples) and text-to-video models, DSP-FWA consistently achieves the highest performance with an AUC of 97.8% and 99.9%, respectively. Similarly, for attribution in text-to-audio data, all the detection models are successfully able to identify the source of the generation model with comparable performance. From these observations, we note that each generative model introduces unique signatures in their generated output. The detection models pick these signatures for a near-perfect performance on model attribution task.

Table 11: Model attribution on Set B

| Attribute | Models | Accuracy | AUC |
|---|---|---|---|
| Images | F3Net | 0.878 | 0.971 |
| | DSP-FWA | 0.911 | 0.978 |
| | MesoInceptionNet | 0.499 | 0.822 |
| | Xception | 0.889 | 0.972 |
| Video | F3Net | 0.994 | 0.999 |
| | DSP-FWA | 0.998 | 0.999 |
| | MesoInceptionNet | 0.909 | 0.994 |
| | Xception | 0.996 | 0.999 |
| Audio | RawGAT-ST | 0.991 | 0.999 |
| | AASIST | 0.993 | 0.999 |
| | Conformer | 0.989 | 0.998 |
| | SSLModel | 0.989 | 0.999 |

## 5   DISCUSSION AND CONCLUSION

In this paper, we introduce the ILLUSION dataset, a significant step towards a comprehensive, multi-modal deepfake resource. Created using 28 state-of-the-art generative models, ILLUSION provides diverse AI-generated content across image, audio, and video modalities, including both curated real-world deepfakes and synthetic media. This design enables models trained on ILLUSION to learn features that extend beyond synthetic artifacts, enhancing generalization across domains, particularly in multilingual and noisy settings. Results show that detection models trained on ILLUSION outperform those trained on existing datasets when evaluated on unseen generative techniques and real-world forgeries. Designed to aid the development of robust, multi-modal, multilingual detection systems, the ILLUSION dataset reveals several key insights through our analysis:

**Multi-Modal Deepfake Detection:** The high performance of models like DSP-FWA and AASIST on both visual and audio data suggests that current models are effective at detecting deepfakes when trained on data from same distribution. However, the disparity in performance between identity swaps and completely synthetic data indicates that models may be learning to identify artifacts specific to the generation method rather than generalizable features of deepfakes.

**Zero-Day Attack Generalization:** The significant drop in performance when models trained on one set are tested on another accentuates the challenge of zero-day attack detection. This suggests that models are currently not robust against deepfakes generated by unfamiliar methods, highlighting the need for diverse datasets like ILLUSION.

**Generalization on Real-World Multi-Lingual Deepfake Media:** The subpar performance of models trained on identity-swap and synthetic data, when tested on real-world deepfakes across various languages, depicts the necessity of a curated, multi-lingual deepfake dataset. Such a dataset is crucial for enabling models to effectively generalize to the diverse deepfakes encountered in the wild.

**Model Attribution:** The ability of models to identify the source generation technique with high accuracy demonstrates that generative models leave distinct signatures in their outputs. This could have implications for the traceability of deepfakes and the accountability of generative model creators.

The ILLUSION dataset specifically addresses deepfake detection challenges using specialized generative AI techniques, while acknowledging that generalized forgery methods—such as digital watermarking, image manipulations, and Photoshop-based edits—represent another important aspect of media forensics. These generalized methods, often easier for deep-learning-based detectors to identify and fundamentally different from generative deepfake techniques, would require additional design considerations to maintain dataset consistency. Moreover, ILLUSION prioritizes quality and diversity over sheer size, with each subset carefully curated for distinct purposes, such as evaluating generalizability or robustness to compression artifacts. By incorporating 28 distinct generative methods, along with multi-modal, multi-lingual, and real-world samples, the dataset minimizes redundancy, ensuring relevance and providing valuable insights into detection performance under diverse conditions. Future extensions of ILLUSION will explore the integration of generalized forgery methods to broaden its scope and utility.

## 6   ETHICS STATEMENT

Our analysis estimates that approximately 245 kg $CO_2$-equivalent was emitted during the creation of this dataset (ML $CO_2$ Impact). Despite this environmental impact, the societal benefits are significant. ILLUSION offers a valuable resource for researchers to explore detection methods across diverse types of fake media. Additionally, its balanced representation of gender and skin tone promotes fairness in the development and evaluation of detection techniques. As a comprehensive multi-modal, multi-lingual deepfake dataset, ILLUSION is instrumental in the global fight against misinformation.

## 7   ACKNOWLEDGEMENT

This research is supported through a grant from MEITY under the National Supercomputing Mission. Thakral received partial support from the PMRF Fellowship and Vatsa is partially supported by the Swarnajayanti Fellowship.

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

# APPENDIX

## A    EXTENDED DETAILS OF ILLUSION DATASET

**Details of Skin Tone Scales:** In this dataset, we have adopted the simplified Fitzpatrick Scale. This scale has been reorganized from six categories to four. The details of each category are as follows:

- Bin 1: Light (Fitzpatrick I and II)
- Bin 2: Light-Medium (Fitzpatrick III)
- Bin 3: Medium-Dark (Fitzpatrick IV)
- Bin 4: Dark (Fitzpatrick V and VI)

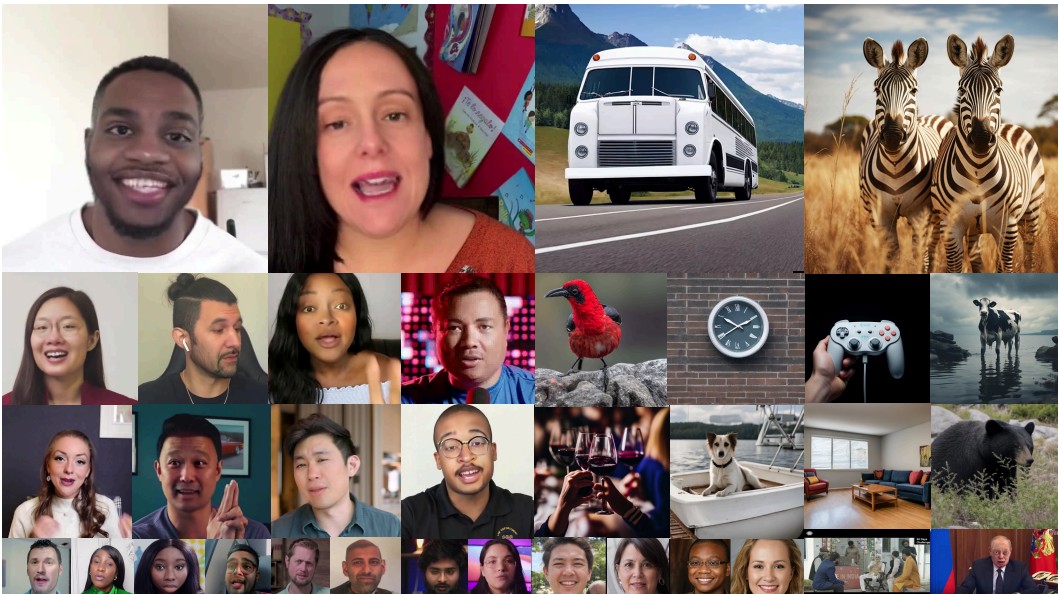

Figure 5: Visualization of diverse samples from the proposed ILLUSION dataset, showcasing variations in resolution, quality, and generative techniques across different sets.

**Performance on Noisy data and Neural Compression:** We investigate the robustness of deep learning models against Gaussian and salt-and-pepper noise, as well as neural compression algorithms. To achieve this, we evaluate baseline models trained on Sets A and B from the proposed ILLUSION dataset. Specifically, we analyze the performance of these models on the noisy and compressed samples from Set C, as summarized in Table 12. Our findings reveal that the models face challenges in generalizing effectively when confronted with noisy and neurally compressed data from Set C.

**Performance on Compressed Data:** In this research, we examine the influence of varying degrees of compression on the performance of visual deepfake detection models. These models are individually trained and evaluated on three distinct versions of dataset A, namely raw, C23, and C40 compressions. Our observations, as detailed in Table 13, reveal that the DSP-FWA model consistently outperforms others across nearly all combinations of training and testing conditions. Interestingly, we notice a decline in performance when the models are trained on high-quality (raw) data and subsequently tested on compressed samples. This suggests a potential challenge in the model's ability to generalize from high-quality training data to lower-quality testing scenarios.

However, promising results emerge when we consider models trained on C23 and C40 compressions. These models demonstrate enhanced generalization when evaluated on both raw and C40 samples, underscoring the benefits of training on compressed data. Consequently, this approach improves the robustness and adaptability of deepfake detection models across varying input qualities.

Table 12: Classification performance on different noise and neural compression added to Set C. The models are evaluated on Gaussian noise (with deviations of 0.5 and 0.75), Salt and Pepper Noise (with noise density of 100 and 150), and Neural Compression (Cheng et al. (2020) and Minnen et al. (2018)). The baseline models are trained on Set A and Set B and tested on the noisy version of Set C.

| Trained On | Models | Gaussian Noise | | Salt and Pepper Noise | | Neural Compression | |
|---|---|---|---|---|---|---|---|
| | | $\sigma = 0.5$ | $\sigma = 0.75$ | 100 | 150 | Cheng et al. | Minnnen et al. |
| Set A | F3Net | 0.612 | 0.601 | 0.634 | 0.622 | 0.525 | 0.481 |
| | DSP-FWA | 0.592 | 0.571 | 0.601 | 0.584 | 0.516 | 0.508 |
| | MesoInceptionNet | 0.553 | 0.52 | 0.538 | 0.495 | 0.502 | 0.497 |
| | Xception | 0.627 | 0.605 | 0.643 | 0.626 | 0.582 | 0.554 |
| Set B | F3Net | 0.604 | 0.593 | 0.627 | 0.606 | 0.544 | 0.504 |
| | DSP-FWA | 0.582 | 0.572 | 0.575 | 0.572 | 0.485 | 0.493 |
| | MesoInceptionNet | 0.537 | 0.544 | 0.513 | 0.508 | 0.498 | 0.498 |
| | Xception | 0.601 | 0.593 | 0.612 | 0.582 | 0.604 | 0.582 |

Table 13: Classification performance for visual components of the dataset for intra and cross-compression of set A

| Trained On | Models | Raw | | | | C-23 | | | | C-40 | | | |
|---|---|---|---|---|---|---|---|---|---|---|---|---|---|
| | | Accuracy | Accuracy-Fake | Accuracy-Real | AUC | Accuracy | Accuracy-Fake | Accuracy-Real | AUC | Accuracy | Accuracy-Fake | Accuracy-Real | AUC |
| Raw | F3Net | 0.851 | 0.751 | 0.951 | 0.945 | 0.826 | 0.703 | 0.940 | 0.927 | 0.775 | 0.734 | 0.813 | 0.852 |
| | DSP-FWA | 0.984 | 0.982 | 0.986 | 0.997 | 0.978 | 0.975 | 0.981 | 0.997 | 0.815 | 0.688 | 0.935 | 0.874 |
| | MesoInceptionNet | 0.505 | 0.999 | 0.882 | 0.883 | 0.481 | 0.999 | 0.001 | 0.849 | 0.486 | 1.000 | 0.000 | 0.730 |
| | Xception | 0.851 | 0.846 | 0.856 | 0.920 | 0.836 | 0.826 | 0.845 | 0.906 | 0.745 | 0.861 | 0.634 | 0.843 |
| C-23 | F3Net | 0.876 | 0.878 | 0.873 | 0.949 | 0.861 | 0.861 | 0.861 | 0.938 | 0.757 | 0.867 | 0.653 | 0.856 |
| | DSP-FWA | 0.979 | 0.987 | 0.971 | 0.995 | 0.977 | 0.986 | 0.968 | 0.996 | 0.810 | 0.931 | 0.696 | 0.930 |
| | MesoInceptionNet | 0.845 | 0.812 | 0.878 | 0.917 | 0.835 | 0.795 | 0.873 | 0.911 | 0.742 | 0.805 | 0.682 | 0.825 |
| | Xception | 0.866 | 0.864 | 0.867 | 0.929 | 0.852 | 0.845 | 0.860 | 0.917 | 0.773 | 0.870 | 0.680 | 0.859 |
| C-40 | F3Net | 0.717 | 0.474 | 0.964 | 0.840 | 0.763 | 0.839 | 0.694 | 0.862 | 0.794 | 0.677 | 0.905 | 0.884 |
| | DSP-FWA | 0.776 | 0.853 | 0.698 | 0.869 | 0.728 | 0.482 | 0.956 | 0.838 | 0.927 | 0.951 | 0.904 | 0.978 |
| | MesoInceptionNet | 0.512 | 0.033 | 0.996 | 0.738 | 0.535 | 0.036 | 0.996 | 0.814 | 0.548 | 0.089 | 0.982 | 0.869 |
| | Xception | 0.754 | 0.582 | 0.927 | 0.847 | 0.768 | 0.606 | 0.918 | 0.851 | 0.809 | 0.818 | 0.799 | 0.879 |

**Comparison with Existing Deepfake Datasets:** We evaluate ILLUSION's uniqueness and relevance in comparison to existing datasets through a series of analyses.

We employed the **Jaccard Similarity Index**, defined as the ratio of the intersection to the union of two sets, to quantitatively assess the overlap between the **ILLUSION** dataset and existing datasets. This metric, ranging from 0 (disjoint sets) to 1 (identical sets), provides a measure of set similarity. The results, as summarized in Table 3, demonstrate a substantial lack of overlap. This confirms ILLUSION's distinctive coverage and its unique contribution to the landscape of generative method datasets.

Next, we utilized the **UpSet plot** (visualized in Figure 6) to analyze the distribution of generative techniques, which revealed both unique contributions and shared intersections. The largest intersection corresponds to generative techniques exclusive to ILLUSION, including diffusion-based algorithms such as DiffSwap, DiffFace, DiffVC, and DiffGAN-TTS, as well as multi-modal synthesis methods. These highlight ILLUSION's novel contributions to media forensics. In contrast, smaller intersections with large-scale datasets like ForgeryNet and KoDF represent shared techniques such as GAN-based image synthesis and lip-sync forgeries, ensuring ILLUSION's benchmarking relevance against state-of-the-art methods. The relatively smaller shared intersections, compared to ILLUSION's unique contributions, highlights the dataset's comprehensive design in bridging gaps across modalities while introducing underexplored generative methods.

To directly assess feature differences, we conducted feature embedding analyses using ResNet50 pre-trained on ImageNet. Uniformly sampled data (50,000 samples each) from ILLUSION and FakeAVCeleb were compared. The t-SNE plot (visualized in Figure 7) reveals distinct clustering patterns with minimal overlap, indicating significant differences in feature distributions. Additionally, the KL divergence of 1.62 between the feature embeddings quantitatively reinforces this dissimilarity. These objective metrics affirm that ILLUSION introduces novel and diverse generative techniques to the detection ecosystem, complementing existing datasets.

# B  GENERATION PIPELINE

In this section, we discuss the methods used for the generation of the proposed ILLUSION dataset.

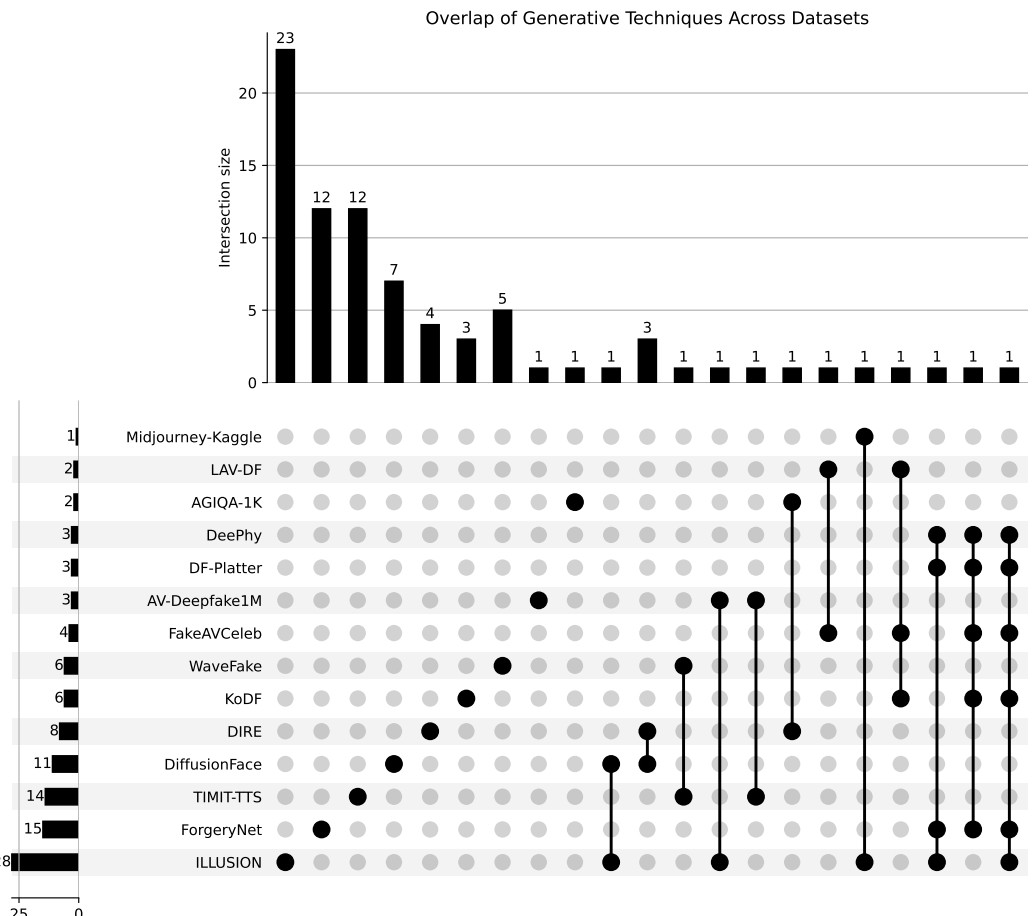

Figure 6: Overlap of Generative Techniques Across Datasets (UpSet Plot) - This plot illustrates the overlap of generative techniques across ILLUSION and other publicly available datasets. The largest intersection corresponds to 23 generative techniques unique to ILLUSION, including diffusion-based methods and advanced multi-modal synthesis techniques. Smaller intersections highlight shared generative methods between ILLUSION and datasets like ForgeryNet, KoDF, and DF-Platter. The plot highlights ILLUSION's distinctiveness while maintaining relevance through its integration of shared techniques.

## B.1 Set A: Identity Swap Deepfakes

The process of generating identity swaps of set A is visualized in Figure 2 (of the main paper). To generate these swaps, we utilize the YouTube videos from the CelebV-Text dataset (Li et al., 2020). We select 200 unique real identities, ensuring that 25 identities were chosen for each sub-group within the four skin-tone bins (as described in section A) and for both sexes, Male and Female, to maintain balance in the dataset in terms of sex and skin-tone group. Each real video clip is 20 seconds long and includes various occlusions, such as hats, glasses, excessive body movements, and harsh lighting, to ensure diversity in our dataset. Furthermore, for Females from the third bin of the Fitzpatrick scale, we scraped 25 videos from YouTube to compensate for the lack of English language videos in CelebV-Text and generate 25 images from "This Person Does Not Exist" (TPDNE) for face-swapping. We then assigned one identity from the YouTube video to another identity from the same generated images.

We utilize 6 video generative models, 7 audio generative models, and an audio-video synchronization tool. Each of them is discussed below in detail.

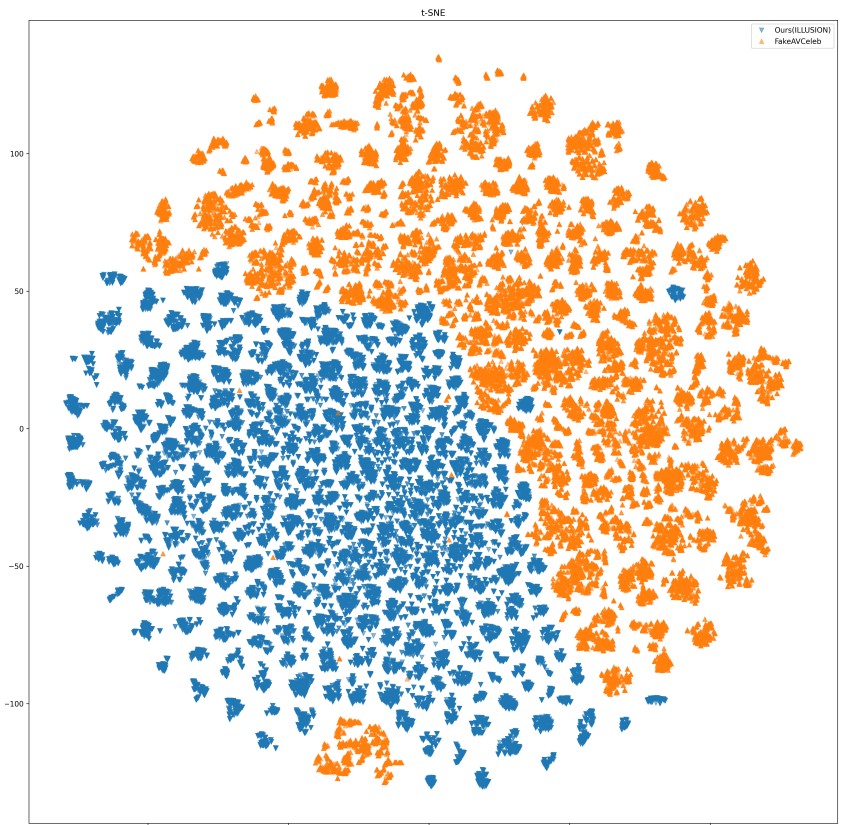

Figure 7: Feature-Level Comparison Between ILLUSION and FakeAVCeleb (PCA and t-SNE) - The PCA (left) and t-SNE (right) plots compare the feature embeddings of 10,000 randomly sampled data points from ILLUSION and FakeAVCeleb, extracted using a ResNet50 model pre-trained on ImageNet. The plots reveal distinct clustering patterns with minimal overlap, indicating significant differences in feature distributions. A KL-divergence of 1.23 further quantifies these differences, validating that ILLUSION provides unique feature representations compared to existing datasets.

**Visual Generation:** For identity swapping i.e., face swapping in videos, 6 different models are employed, namely, MobileFaceSwap (Xu et al., 2022b), FSGAN (Nirkin et al., 2019), FaceShifters (Li et al., 2019b), ROOP (Roop, 2023), DiffFace (Kim et al., 2022), and DiffSwap (Zhao et al., 2023). For all possible combinations within a sub-group, the source (from which facial features are extracted) is swapped onto the target faces (which provide the face shape and background). Each manipulated video has the same duration as the corresponding real video in the base dataset. Each of the video models used for manipulating faces are discussed below:

- **FSGAN (Nirkin et al., 2019)** is a Face Swapping GAN model for face swapping and reenactment. It is subject-agnostic and can be applied to pairs of faces without requiring training on those faces. The model for video-to-video is used with default hyperparameters.

- **FaceShifter (Li et al., 2019b)** generates the swapped face, extracting target attributes and embedding them onto the source. FaceShifter is robust against occlusions using two network pipelines, AEI-Net and HEAR-Net, trained to recover anomaly regions self-supervised without manual annotations. The pre-trained weights of ArcFace and AEI-Net are used to extract identity embeddings, and utilize source images and target videos to generate videos.

- **MobileFaceSwap (Xu et al., 2022b)** is a lightweight Identity-aware Dynamic Network (IDN) for subject-agnostic face swapping by dynamically adjusting the model parameters according to the identity information. The ArcFace and model pre-trained weights are used for identity embedding extraction and inference. We utilize source images and target videos to generate videos with default hyperparameters.

- **Roop (Roop, 2023)** is one of the most widely used open-source deepfake generation software. Roop uses a face-swapping pre-trained model of InsightFace by DeepInsight. The default hyperparameters are used for generation while extracting a source frame to swap onto a target video.
- **DiffFace (Kim et al., 2022)** is a diffusion-based face-swapping framework composed of training ID conditional DDPM, sampling with facial guidance, and target-preserving blending. 40 equidistant frames are extracted from videos to create face swap images for the unseen test set. The default hyperparameters and pre-trained weights are utilized.
- **DiffSwap (Zhao et al., 2023)** is a diffusion model-based framework for high-fidelity and controllable face swapping. We use the pre-trained weights for face and landmark detection.

**Audio Generation:** The identity swapping for audio i.e., audio spoofing is performed using 7 different audio-generative models, namely, FreeVC (Li et al., 2023), XTTS (Eren & The Coqui TTS Team, 2021), DiffVC (Popov et al., 2021b), DiffHierVC (Choi et al., 2023), YourTTS (Casanova et al., 2022), DiffGAN-TTS (Liu et al., 2022), GradTTS (Popov et al., 2021a) to create this dataset. The audio is extracted from the selected videos, and the MMS model (Pratap et al., 2024) is used to identify the language and English audio is selected. Next, we generate a transcription of the audio using the MMS model. This generated transcription is then used to generate synthetic audio. Additionally, the raw audio is used to create voice-cloned audio of the source speaker. We clone voices for all the combinations of source (speech which is cloned) and target (text of the audio borrowed from) within a sub-group. This ensures that the dataset's quality is maintained and that altered content is realistic by preserving the unique accents within a sub-group. Each audio is limited to 20 seconds in length to maintain the quality of the synthesized speech without compromising on the audio's duration. We discuss each of the audio models used for generating audio below:

- **FreeVC (Li et al., 2023)** is an end-to-end model for voice conversion (VC) based on the approach proposed in VITS. The FreeVC model is a variational auto-encoder-based architecture pre-trained on the VCTK speech corpus. We use the model provided by Coqui to generate cloned voices using the pre-trained speaker encoder model.
- **XTTS (Eren & The Coqui TTS Team, 2021)** is an end-to-end voice generation model for voice cloning into different languages built on Tortoise. XTTS has significant model changes that simplify cross-language voice cloning and multi-lingual speech generation. Coqui is used to generate clones.
- **DiffVC (Popov et al., 2021b)** is another zero-shot many-to-many voice conversion method designed for the general case when source and target speakers do not belong to the training dataset. The model is trained using HifiGAN vocoder, and for voice conversion, the model is trained on VCTK and LibriTTS speech corpus. We use the pre-trained speaker encoder and voice conversion model trained on VCTK for cloning.
- **Diff-HierVC (Choi et al., 2023)** is a diffusion-based hierarchical VC system for voice conversion. For voice style transfer, Diff-HierVC uses two diffusion models, DiffVoice and DiffPitch. We use BigVGAN vocoder and pre-trained model weights with default hyperparameters.
- **YourTTS (Casanova et al., 2022)** was originally a text-to-speech model based on the end-to-end VITS but with a separate speaker embedding (from a speaker recognition model) to encode speaker identity. The inclusion of speaker encoding allows the use of YourTTS in a zero-shot voice conversion manner by simply substituting the embedding of one speaker with the embedding of another. Coqui implementation of the model is used for generation.
- **DiffGAN-TTS (Liu et al., 2022)** is a Denoising Diffusion Probabilistic Model (DDPM) - based TTS model. It is based on denoising diffusion generative adversarial networks (GANs), which adopt an adversarially-trained expressive model to approximate the denoising distribution. The shallow model trained on the VCTK dataset for 400000 steps is used. We assign a unique speaker ID from VCTK data to an identity of a sub-group and create spoofs within a train or test set of sub-groups.
- **GradTTS (Popov et al., 2021a)** is a text-to-speech model with a score-based decoder producing mel-spectrograms by gradually transforming noise predicted by the encoder and aligning with text input employing Monotonic Alignment Search. GradTTS uses a HifiGAN

vocoder. We use the pre-trained weights of the model and vocoder trained on the LibriTTS dataset. A unique speaker ID from VCTK data is assigned to an identity in a sub-group, and spoofs are created within a train or test set of sub-groups.

**Audio-Video Synchronization:** We combine the audio and video using the Audalign (Alignment). It is an open-source project that uses audio fingerprinting, cross-correlation, cross-correlation with spectrograms, or visual alignment techniques to align audio and video timestamps. The extracted audio from the real video is combined with face-swapped manipulated videos, giving us our dataset's ii) Real Audio and Fake Video class. Moreover, we also combined the generated audio spoofs after syncing with real and face-swapped video. This gives us the two other classes of our dataset: iii) Fake Audio and Real Video and iv) Fake Audio and Fake Video. This gives us all four classes of the generated dataset.

## B.2 SET B: AI GENERATED CONTENT

With the advent of generative AI methods, the amount of completely synthetic media across modalities has surged in social media platforms. These methods better capture contextual information since they generate media based on a textual prompt or guidance from pre-existing media. In this set, we present a subset of completely synthetic AI-generated content. The generation procedure for each modality is discussed below in detail-

**Video Generation:** We use the MSRVTT (Xu et al., 2016) dataset for video-caption pairs. It is a large-scale dataset for open-domain video captioning containing video clips annotated with 20 English sentences by Amazon Mechanical Turks. From the publicly available dataset, 7010 captions are used to generate videos. For this, we use three different text-to-video models: Text2Video-Zero (Khachatryan et al., 2023), ModelScopeT2V (Wang et al., 2023a) and ZeroScope (Zeroscope). 24 frames are generated for every video from Text2Video-Zero and ZeroScope, and 40 frames are generated for each from ModelScopeT2V. More frames are generated from ModelScopeT2V because the videos generated from this model have a Shutterstock watermark. The higher number of frames can separately be used later to check how different detection algorithms perform on samples that contain a watermark. Watermarks have become increasingly common in internet content as more organizations prefer to add their watermark on their respective media. Each of the generative video models is discussed briefly below:

- **ModelScopeT2V (Wang et al., 2023a)** is a generative text-to-video synthesis model that evolves from Stable Diffusion. It uses spatiotemporal blocks, VQGAN, a text encoder, and a denoising UNet, resulting in consistent frame generation and smooth movement transitions. The videos generated from this model have a Shutterstock watermark. We use its public implementation, which is available in the Huggingface diffusers library.

- **Zeroscope (Zeroscope)** is a generative model based on ModelScope itself, but it has been improved to produce higher-quality videos with a fixed aspect ratio and no Shutterstock watermark. We used zeroscape-v2-576w.

- **Text2Video-Zero (Khachatryan et al., 2023)** is a zero-shot text-to-video generation model. It uses existing text-to-image synthesis methods (e.g., Stable Diffusion), making them suitable for the video domain. Mainly, it includes frame-level self-attention using new cross-frame attention of each frame on the first frame to preserve the context, appearance, and identity of the foreground object.

**Audio Generation:** We use the MusicCaps (Agostinelli et al., 2023) dataset to generate music samples, a large-scale dataset containing 5,211 music examples, each labelled with an English aspect list and a free text caption written by musicians. For each caption, we use three text-to-music models, namely: AudioLDM (Liu et al., 2023), MusicGen (Copet et al., 2023) and MAGNeT (Ziv et al., 2024). Further, we download the original audio clip from Youtube, and corresponding to each audio clip, we use an audio-to-audio model: Audio Diffusion (Face). Each of the generative models are discussed below:

- **MusicGen (Copet et al., 2023)** is an end-to-end model for text-to-audio generation specifically tailored for music generation. It comprises a single-stage transformer language

model capable of generating conditioned high-quality music samples. We use its public implementation from the AudioCraft library.

- **MAGNeT (Ziv et al., 2024)** is an end-to-end model for text-conditioned music generation. It is a Masked Audio Generation method utilizing a single non-autoregressive transformer. Its public implementation from the AudioCraft library is used.

- **AudioLDM (Liu et al., 2023)** is an end-to-end model for text conditioned audio generation. A latent diffusion model (LDM) learns continuous audio representations from CLAP latent. We used its public implementation from the diffuser library.

- **Audio Diffusion (Face)** is an end-to-end model for audio to audio generation. It converts audio input into a mel spectrogram, then uses a Variational Auto-Encoder (VAE) model to encode and decode images to and from latent, and finally a UNet2DConditionModel to denoise the encoded image latents. Public implementation from the diffusers library is used for generation.

**Image Generation:** We use the training set of the COCO dataset to get 118,287 image-caption pairs. Now, corresponding to each caption in the dataset, we use four text-to-image models, namely: Stable Diffusion XL (Podell et al., 2023), Kandinsky 2.1 (Razzhigaev et al., 2023), MultiDiffusion (Bar-Tal et al., 2023) and SDXL-Turbo (Sauer et al., 2023). Further, the original image from the dataset is kept as the corresponding real for each generated image. We discuss each of the generative models below:

- **Stable Diffusion XL (Podell et al., 2023)** is an end-to-end model for text-conditioned image generation and is an improvement to the previous stable diffusion models with a much larger UNet and a combination of OpenCLIP ViT-bigG/14 text encoder with the original text encoder. We used its public implementation from the diffusers library.

- **Kandinsky 2.1 (Razzhigaev et al., 2023)** is an end-to-end model for text-conditioned image generation. As the text and image encoder, it uses the CLIP model and diffusion image prior (mapping) between latent spaces of CLIP modalities to generate photo-realistic images. We use its public implementation from the diffusers library.

- **MultiDiffusion (Bar-Tal et al., 2023)** is a recent technique for text-conditioned image generation, making any pre-trained text-to-image model usable for generation without fine-tuning and training with a higher control over output. We used it in conjugation with StableDiffusionPanoramaPipeline as available in its public implementation from the diffusers library.

- **SDXL-Turbo (Sauer et al., 2023)** is an end-to-end model for text to image generation. It is a distilled version of SDXL-1.0 and uses a novel training method called Adversarial Diffusion Distillation (ADD) for much faster, high-quality generation. We used its public implementation from the diffusers library.

**Synthetic-Faces Dataset:** We scrape image samples from a popular deep fake generation website `thispersondoesnotexist.com`. Further, we classified these samples as per sex and skin tone into eight subgroups to create a demographically balanced dataset that contains 1025 synthetic faces for each subgroup cumulating to a total of 8,200 synthetic face images. Now, to have an equal number of real faces in our dataset, we collect an equal number of samples from Balanced Faces in the Wild (BFW) (Robinson, 2022).

## B.3 SET C: FAKES IN THE WILD

We use Midjourney (Midjourney) and Artguru (ArtGuru), both premium tools that allow users to generate synthetic images via text-to-image modality through a premium API and a web interface, respectively. For Midjourney, we use prompts from the validation set of the COCO dataset, and for each prompt, we get four corresponding images to generate a total of 20,000 images. Artguru, however, specializes more in generating identity-driven images for a given prompt. Therefore, we randomly pick 1000 prompts from the person class of the COCO dataset to generate a total of 1000 images. Apart from these images, we have 24 real-world doctored videos used for testing.

## C    BASELINE MODELS

In our evaluation of the ILLUSION dataset, we employ a diverse set of eleven deepfake detection methods. The selection of these methods is primarily guided by the availability of their respective codebases. For each method, we adhere to the default parameters provided, ensuring a fair and consistent comparison across all detection methods. In the following sections, we provide a succinct discussion of each of these detection methods:

### C.1    VISUAL MODELS

- **MesoInceptionNet (Afchar et al., 2018)** is a CNN architecture used for deepfake and face-to-face image forgery detection. It develops on the Inception module. Going deeper with convolutions for the first two convolutional layers of the Meso-4 architecture, MesoInception-4 is created.

- **XceptionNet (Chollet, 2017)** is a deep CNN architecture using depthwise separable convolutions. Inception modules in CNNs can be seen as a transitional stage between conventional convolutions and depthwise separable convolutions, which combine depthwise and point-wise convolutions. This insight leads to a new Inception-inspired deep CNN design, where depthwise separable convolutions replace Inception modules.

- **DSP-FWA (Li & Lyu, 2018)** It uses face-warping artifacts from deepfake generation to identify swapped videos. It employs a CNN trained with negative examples created by mimicking affine face warping steps and positive face images. To simulate artifacts, these negative examples involve recognizing faces, resizing them, applying Gaussian blur, and then affine warping. The technique uses CNN models like VGG16 and ResNet to detect inconsistencies between manipulated facial regions and their surroundings.

- **F3Net (Wei et al., 2020)** or Frequency in Face Forgery Network is a network that deep mines the forgery patterns through a two-stream collaborative learning framework, utilising two complementary but distinct frequency-aware clues: 1) frequency-aware decomposed image components, and 2) local frequency statistics. It uses DCT frequency-domain transformation.

### C.2    AUDIO MODELS

- **RawGAT-ST (Tak et al., 2021)**, is a spectro-temporal graph attention network that learns to detect spoofed speech from raw waveform inputs. It creates separate spectral and temporal sub-graphs, employs graph attention to capture relationships within each sub-graph, fuses the sub-graphs at the model level, and uses graph pooling to improve discrimination. This approach automatically learns relevant spectro-temporal artifacts without hand-crafted features or separate systems for different attacks.

- **AASIST (Jung et al., 2022)** or Attention-Augmented Speaker Inconsistency Stacking Transformer for detecting spoofed utterances. It uses a heterogeneous stacking graph attention layer that models artifacts spanning both temporal and spectral domains using a heterogeneous attention mechanism and a stack node. The model employs a max graph operation with a competitive mechanism and an extended readout scheme.

- **SSLModel (Tak et al., 2022)** proposes an embedding fusion scheme that combines representations from supervised learning (SL) and self-supervised learning (SSL) models for spoofed speech detection. It performs weighted score fusion between SL and best SSL models and then minimizes the embedding distribution between selected SL and SSL representations using a comprehensive statistical analysis to select appropriate layers. This fusion approach leverages knowledge from both SL and SSL frameworks to improve performance.

- **Conformer (Gulati et al., 2020)** model has statistical pooling for jointly performing speaker verification and voice spoofing detection. It utilizes the Conformer architecture, which combines Transformer and CNNs and employs statistical pooling to aggregate frame-level representations. The model is trained on both tasks simultaneously, enabling it to reject both spoofed utterances and utterances from different speakers.

### C.3 Multi-modal Models

- **FACTOR (Reiss et al., 2023)** is a detection method that utilizes fact-checking adapted from fake news detection. It verifies the claimed facts (e.g., identity) by comparing them with the actual observed media (e.g., appearance) to identify real versus fake content. It does not require training, uses readily available features, and achieves higher accuracy without prior exposure to deepfakes, especially in face-swapping and audio-visual synthesis scenarios.

- **MRDF (Zou et al., 2024)** proposes cross-modality and within-modality regularization to maintain modality distinctions during multi-modal representation learning. The approach includes an audio-visual transformer module for modality correspondence and a cross-modality regularization module to align paired audio-visual signals. Additionally, a within-modality regularization module refines unimodal representations to retain modality-specific details.

- **Unimodal Ensembling** integrates an ensemble of two best performing models, F3Net and SSLModel. For any given deepfake input, we first extract the video and audio components. Subsequently, we fuse the predictions derived from each modality to arrive at a final prediction. This process ensures a comprehensive analysis of the deepfake content, leveraging both visual and auditory cues.

