# ILLUSION: UNVEILING TRUTH WITH A COMPREHENSIVE MULTI-MODAL, MULTI-LINGUAL DEEPFAKE DATASET

**Kartik Thakral**[*1], **Rishabh Ranjan**[*1], **Akanksha Singh**[1,2], **Akshat Jain**[1],
**Mayank Vatsa**[1], **and Richa Singh**[1]
[1]IIT Jodhpur, India, [2]IISER Bhopal, India

## 1 LINKS

The ILLUSION Leaderboard for the ICLR 2025 reviewers is available here: `https://www.iab-rubric.org/illusion-database`

## 2 AUTHOR STATEMENT

Our dataset can be commonly copied and redistributed in any medium or format. Our dataset can be accessed using the following license: CC-BY-4.0 (`https://creativecommons.org/licenses/by/4.0/`).

## 3 MOTIVATION

**For what purpose was the dataset created? Was there a specific task in mind? Was there specific gap that needed to be filled? Please provide a description.**

Deepfakes have emerged as a formidable challenge, posing significant technical, social, and ethical dilemmas. The ease with which deepfakes can be generated has given rise to numerous ethical, security, and privacy concerns. Consequently, there is an urgent need to develop robust deepfake detection methods. However, the effectiveness of these methods hinges on the availability of comprehensive deepfake datasets.

While there are existing deepfake datasets, they often lack comprehensiveness, particularly in terms of multimodal deepfakes that encompass a wide spectrum of fakes, ranging from identity swaps to synthetic fakes. In this work, we strive to bridge this gap by introducing a multimodal deepfake dataset. This dataset, synchronized for audio and video, offers significant variations in generation techniques, thereby providing a more robust foundation for deepfake detection research.

**Who created the dataset (e.g., which team, research group) and on behalf of which entity (e.g., company, institution, organization)?**

Only the authors of this paper have contributed towards the creation of the dataset.

**Who funded the creation of the dataset? If there is an associated grant, please provide the name of the grantor and the grant name and number.**

Thakral is partially supported by PMRF Fellowship and Vatsa is partially supported by Swarnjayanti Fellowship.

**Were any ethical review processes conducted (e.g., by an institutional review board)?**

We have obtained the dataset's approval from the Institutional Review Board (IRB).

---

[*]Equal Contribution

## 4 COMPOSITION

**What do the instances that comprise the dataset represent (e.g., documents, photos, people, countries)? Are there multiple types of instances (e.g., movies, users, and ratings; people and interactions between them; nodes and edges)? How many instances are there in total (of each type, if appropriate)?**

The dataset is a comprehensive multimodal deepfake collection. It is crafted using a diverse array of 28 proprietary and open-source generation methods, resulting in an expansive repository of over 1.3 million samples. These samples encompass a wide range of deepfakes, including identity swaps, synthetically fabricated media, synthetic faces, and curated real-world deepfakes. The dataset is partitioned into three distinct sets. Set A is an identity forgery dataset, comprising a mix of 14 models rooted in various technologies such as Generative Adversarial Networks (GANs), Variational Autoencoders (VAEs), transformers, diffusion-based models, and audio synchronization techniques. Set B is a rich collection of AI-generated synthetic data, spanning three media modalities: image, audio, and video. The generation process for this set employed a total of 11 popular, publicly available models for text-to-image, text-to-audio, and text-to-video conversion. Additionally, it includes a subset of entirely synthetic faces, generated using the renowned "This Person Does Not Exist" model. Lastly, Set C serves as a test set and comprises a diverse pool of real-world AI-generated content (AIGC), sampled from a variety of sources. This comprehensive and diverse dataset is designed to provide a robust foundation for deepfake detection research.

**Does the dataset contain all possible instances or is it a sample (not necessarily random) of instances from a larger set?**

Yes, We include all possible instances of our dataset. We plan to release the complete dataset after the paper is accepted.

**What data does each instance consist of? "Raw" data (e.g., unprocessed text or images) or features? In either case, please provide a description.**

The dataset consists of processed videos, audios and images in format '.mp4', '.wav', and '.png', respectively. The set A comprises of three compression factors i.e., raw , C23, and C40.

**Is there a label or target associated with each instance?**

Yes, The dataset contains 'Real' and 'Fake' labels corresponding to each sample.

**Is any information missing from individual instances? If so, please provide a description, explaining why this information is missing (e.g., because it was unavailable). This does not include intentionally removed information, but might include, e.g., redacted text.**

No, all of the annotation information has been provided.

**Are relationships between individual instances made explicit (e.g., users' movie ratings, social network links)? If so, please describe how these relationships are made explicit.**

There is no individual information in the dataset.

**Are there recommended data splits (e.g., training, development/validation, testing)? If so, please provide a description of these splits, explaining the rationale behind them.**

Yes, the dataset contains train and test splits for set A and set B. Set C is essentially a test set only. Each test set contains unseen samples.

**Are there any errors, sources of noise, or redundancies in the dataset? If so, please provide a description.**

The authors have tried their best to manually remove any low-quality or bad samples from the dataset.

**Is the dataset self-contained, or does it link to or otherwise rely on external resources (e.g., websites, tweets, other datasets)? If it links to or relies on external resources,**

Mostly, dataset is self-contained, however, since our real videos are borrowed from different datasets, researcher can refer to them.

**Does the dataset contain data that might be considered confidential (e.g., data that is protected by legal privilege or by doctor patient confidentiality, data that includes the content of individuals' non-public communications)?**

No, the dataset do not contain any information that can be considered confidential.

**Does the dataset contain data that, if viewed directly, might be offensive, insulting, threatening, or might otherwise cause anxiety?**

No, not at all.

**Does the dataset relate to people? If not, you may skip the remaining questions in this section.**

Yes, dataset is realted to people.

**Does the dataset identify any subpopulations (e.g., by age, gender)? If so, please describe how these subpopulations are identified and provide a description of their respective distributions within the dataset.**

Yes, the dataset is balanced with respect to people in accordance with the Fitzpatrick scale and sex.

**Is it possible to identify individuals (i.e., one or more natural persons), either directly or indirectly (i.e., in combination with other data) from the dataset?**

Since some part of the of set A is borrowed from CelebV-Text dataset which is collected from YouTube and set C is curated from the wild, it is possible to identify the people.

**Does the dataset contain data that might be considered sensitive in any way (e.g., data that reveals racial or ethnic origins, sexual orientations, religious beliefs, political opinions or union memberships, or locations; financial or health data; biometric or genetic data; forms of government identification, such as social security numbers; criminal history)?**

No. The dataset does not contain any sensitive information.

**Collection How was the data associated with each instance acquired? Was the data directly observable (e.g., raw text, movie ratings), reported by subjects (e.g., survey responses), or indirectly inferred/derived from other data (e.g., part-of-speech tags, model-based guesses for age or language)? If data was reported by subjects or indirectly inferred/derived from other data, was the data validated/verified? If so, please describe how.**

To generate the fakes, we sourced the real data from multiple open-source datasets like CelebV-Text, MS-COCO, MusicCaps, and MSRVTT. The required data was directly observable and utilized accordingly.

**What mechanisms or procedures were used to collect the data (e.g., hardware apparatus or sensor, manual human curation, software program, software API)? How were these mechanisms or procedures validated?**

Not Applicable: we did not collect any data.

**If the dataset is a sample from a larger set, what was the sampling strategy (e.g., deterministic, probabilistic with specific sampling probabilities)?**

We use a stratified probabilistic sampling technique to sample 25 identities for each sub-group created on the basis of skin tone and sex out from the CelebVText dataset.

**Who was involved in the data collection process (e.g., students, crowdworkers, contractors) and how were they compensated (e.g., how much were crowdworkers paid)?**

We did not collect any data.

**Over what timeframe was the data collected? Does this timeframe match the creation timeframe of the data associated with the instances (e.g., recent crawl of old news articles)? If not, please describe the timeframe in which the data associated with the instances was created.**

We did not collect any datasets from individuals. Instead, we use the already published CelebV-Text, MS-COCO, MusicCaps, and MSRVTT datasets as our real counterpart corpus.

**Were any ethical review processes conducted (e.g., by an institutional review board)? If so, please provide a description of these review processes, including the outcomes, as well as a link or other access point to any supporting documentation.**

We have obtained permission from the Institutional Review Board (IRB) for the proposed dataset.

**Does the dataset relate to people? If not, you may skip the remainder of the questions in this section.**

Yes, the dataset includes deepfake videos featuring individuals.

**Did you collect the data from the individuals in question directly, or obtain it via third parties or other sources (e.g., websites)?**

We do not collect any data from individuals. However, we use real videos from the CelebV-Text dataset, which are already publicly available.

**Did the individuals in question consent to the collection and use of their data? If so, please describe (or show with screenshots or other information) how consent was requested and provided, and provide a link or other access point to, or otherwise reproduce, the exact language to which the individuals consented.**

Not Applicable, we do not collect any data from individuals. **If consent was obtained, were the consenting individuals provided with a mechanism to revoke their consent in the future or for certain uses? If so, please provide a description, as well as a link or other access point to the mechanism (if appropriate).**

Not Applicable

**Has an analysis of the potential impact of the dataset and its use on data subjects (e.g., a data protection impact analysis) been conducted? If so, please provide a description of this analysis, including the outcomes, as well as a link or other access point to any supporting documentation.**

Not Applicable

## 5 PREPROCESSING, CLEANING AND LABELING

**Was any preprocessing/cleaning/labeling of the data done (e.g., discretization or bucketing, tokenization, part-of-speech tagging, SIFT feature extraction, removal of instances, processing of missing values)? If so, please provide a description. If not, you may skip the remainder of the questions in this section.**

From the CelebV-Text dataset, identities were segregated based on skin tone using the Fitzpatrick Scale and sex into sub-groups. Videos with only one person in the frame were selected. Variations in occlusions, lighting, background, etc, have been taken into consideration. Next, the MMS model was used for language identification, and English videos were selected from it. Videos were trimmed to 20 seconds in length, maintaining the $512 \times 512$ resolution and original fps. Frames and audio were extracted from each video using the FFmpeg library for the down-the-line generation process. Furthermore, the MMS model was used for transcription.

**Was the "raw" data saved in addition to the preprocessed/cleaned/labeled data (e.g., to support unanticipated future uses)? If so, please provide a link or other access point to the "raw" data.**

No

**Is the software used to preprocess/clean/label the instances available? If so, please provide a link or other access point.**

We use Python libraries/packages to process the dataset.

## 6 USE CASES

**Has the dataset been used for any tasks already? If so, please provide a description.**

The dataset can be used for uni-modal, multi-modal deepfake detection. The dataset can also be used for model attribution tasks. We show the results of these tasks in the main paper.

**Is there a repository that links to any or all papers or systems that use the dataset? If so, please provide a link or other access point. What (other) tasks could the dataset be used for?**

We show the performance of different baseline models on our proposed dataset. All the details are available in the main paper and supplementary file.

**Is there anything about the composition of the dataset or the way it was collected and preprocessed/cleaned/labeled that might impact future uses?**

No

**Are there tasks for which the dataset should not be used? If so, please provide a description.**

The tasks related to deepfake attacks on secure systems are strictly prohibited. We carefully monitor dataset requests and only approve releases for academic purposes.

## 7  DISTRIBUTION

**Will the dataset be distributed to third parties outside of the entity (e.g., company, institution, organization) on behalf of which the dataset was created? If so, please provide a description.**

The dataset will be available for research to academic institutions after signing the license form. It will not be accessible for commercial use.

**How will the dataset will be distributed (e.g., tarball on website, API, GitHub)? Does the dataset have a digital object identifier (DOI)?**

The dataset will be made publicly available for research only. We will release the dataset only for academic purposes after signing a licence form, which will help restrict the use of the dataset by malicious actors.

**When will the dataset be distributed?**

We will host the dataset after the paper is accepted.

**Will the dataset be distributed under a copyright or other intellectual property (IP) license, and/or under applicable terms of use (ToU)? If so, please describe this license and/or ToU, and provide a link or other access point to, or otherwise reproduce, any relevant licensing terms or ToU, as well as any fees associated with these restrictions.**

The ILLUSION dataset will be available under the Creative Commons 4.0 license.

**Have any third parties imposed IP-based or other restrictions on the data associated with the instances? If so, please describe these restrictions, and provide a link or other access point to, or otherwise reproduce, any relevant licensing terms, as well as any fees associated with these restrictions.**

The restrictions on the borrowed dataset are limited to commercial-based distribution and are free to use for research purposes. Keeping these restrictions in mind, we will release our dataset only for academic purposes.

**Do any export controls or other regulatory restrictions apply to the dataset or to individual instances? If so, please describe these restrictions, and provide a link or other access point to, or otherwise reproduce, any supporting documentation.**

Not Applicable

## 8  MAINTENANCE

**Who is supporting/hosting/maintaining the dataset? How can the owner/curator/manager of the dataset be contacted (e.g., email address)?**

The authors are responsible for maintaining the dataset, which will be hosted on our website.

**Is there an erratum? If so, please provide a link or other access point.**

No

**Will the dataset be updated (e.g., to correct labeling errors, add new instances, delete instances)? If so, please describe how often, by whom, and how updates will be communicated to users (e.g., mailing list, GitHub)?**

If there are any issues with our dataset, we will promptly update the information and highlight it on our dataset website.

**If the dataset relates to people, are there applicable limits on the retention of the data associated with the instances (e.g., were individuals in question told that their data would be retained for a fixed period of time and then deleted)? If so, please describe these limits and explain how they will be enforced.**

There are no applicable limits on the retention of the dataset.

**Will older versions of the dataset continue to be supported/hosted/maintained? If so, please describe how. If not, please describe how its obsolescence will be communicated to users.**

There is no older version to this dataset.

**If others want to extend/augment/build on/contribute to the dataset, is there a mechanism for them to do so? If so, please provide a description. Will these contributions be validated/verified? If so, please describe how. If not, why not? Is there a process for communicating/distributing these contributions to other users? If so, please provide a description.**

Others can contact us to propose fixes and extensions. We welcome the opportunity to expand our work.