# OpenReview forum: "ILLUSION: Unveiling Truth with a Comprehensive Multi-Modal, Multi-Lingual Deepfake Dataset"
_ICLR.cc/2025/Conference — ICLR 2025 Poster_

### Official Review · Reviewer_pXKi · 2024-10-19

**Soundness:** 2
**Presentation:** 3
**Contribution:** 2
**Rating:** 5
**Confidence:** 5

**Summary:**

In this paper, the authors introduce a large-scale, comprehensive deepfake dataset containing nearly 30 different forgery methods, divided into four subsets. The dataset includes multi-modal and multi-lingual fake videos, with a balanced distribution across gender and age groups. The authors also conducted exploratory benchmarking on their dataset and highlighted the challenges that existing detectors face when identifying deepfakes in real-world scenarios.

**Strengths:**

- A large-scale dataset containing various forgery methods is proposed.

- A benchmark for the proposed dataset is constructed.

**Weaknesses:**

- Uniqueness of the dataset. As a large-scale multimodal dataset, ILLUSION does not exhibit particularly distinctive features. Previous datasets, such as ForgeryNet [A] and KoDF [B], have already focused on large-scale video and image forgeries. Additionally, many recent datasets have been proposed for the detection of AIGC content [C][D][E][F][G]. While the integration of various forgery methods is certainly commendable and meaningful, further discussion is needed on how this integration can contribute to advancing research.

- The comparison with related datasets is insufficient. The authors should enhance Figure 1 by including, but not limited to, datasets in [A]-[G].

- The authors are advised to include more state-of-the-art deepfake detectors to create a more comprehensive benchmark.

[A] ForgeryNet: A Versatile Benchmark for Comprehensive Forgery Analysis
[B] KoDF: A Large-scale Korean DeepFake Detection Dataset
[C] DIRE for Diffusion-Generated Image Detection
[D] Diffusion Facial Forgery Detection
[E] DiffusionFace: Towards a Comprehensive Dataset for Diffusion-Based Face Forgery Analysis
[F] GenImage: A Million-Scale Benchmark for Detecting AI-Generated Image
[G] On The Detection of Synthetic Images Generated by Diffusion Models

**Questions:**

Please see Weaknesses

---

> ### Author Response · Authors · 2024-11-28
>
> We sincerely thank you for your feedback and for recognizing the contributions of our work in proposing a large-scale dataset containing various forgery methods and constructing a benchmark. Below, we present our responses and clarifications to your feedback:
>
> ### 1. Uniqueness of the Dataset and Comparison with Existing Datasets
> We appreciate your observation and would like to highlight the unique aspects of the *ILLUSION* dataset that distinguish it from existing datasets:
>
> - **Comprehensive Multi-Modality**:   *ILLUSION* is the first dataset to integrate audio, video, and image modalities in a single, cohesive resource. While datasets like ForgeryNet [A] and KoDF [B] primarily focus on image and video forgeries, *ILLUSION* includes synchronized audio-video forgeries and standalone audio manipulations. This multi-modal approach is crucial for developing detection methods capable of handling complex, cross-modal deepfakes that are increasingly prevalent in real-world scenarios.
>
> - **Multi-Lingual Diversity**:   Our dataset spans 26 languages, significantly expanding linguistic diversity compared to existing datasets that are predominantly monolingual or focus on a single language (e.g., KoDF focuses on Korean). This feature allows researchers to evaluate and improve detection models in multi-lingual contexts, addressing a critical gap where language variations can impact detection performance.
>
> - **Diverse Generative Techniques**:   *ILLUSION* encompasses 28 state-of-the-art generative methods, including diffusion models, voice cloning, lip-sync forgeries, and GAN-based image synthesis. This diversity provides a more comprehensive representation of the spectrum of deepfake generation techniques, enabling the development of detection models that can generalize across different types of manipulations. Furthermore, *ILLUSION* is the first dataset to incorporate diffusion-based generation algorithms—such as DiffSwap, DiffFace, DiffVC, DiffHierVC, and DiffGAN-TTS—for creating multi-modal deepfakes. The inclusion of these unique models introduces distinct digital fingerprints not present in previous resources.
>
> - **Real-World Forgeries and Compression Artifacts**:   We include curated real-world deepfakes sourced from social media and other platforms, as well as samples with various compression levels and noise artifacts. This enhances the dataset's practicality by simulating challenges faced in real-world detection, such as quality degradation and post-processing effects.
>
> - **Structured Evaluation Protocols**:   We provide detailed evaluation protocols organized into multiple settings, such as cross-modal detection, robustness to compression, zero-day attack scenarios, and cross-dataset generalization. This structured approach facilitates comprehensive benchmarking and helps identify specific areas where detection methods need improvement.
>
> We have updated Section 6 to discuss the following possible directions:
> - Enabling the development of robust, multi-modal, and multi-lingual deepfake detection systems.
> - Providing a platform to evaluate models under realistic conditions that closely mimic real-world challenges.
> - Highlighting the limitations of current detection methods, thereby guiding future research directions.
>
> We believe that *ILLUSION* fills a critical gap in the current landscape of deepfake detection datasets and serves as a valuable resource for the research community.
>
> ### 2. Inclusion of More State-of-the-Art Detectors
> To address the dynamic landscape of deepfake detection algorithms and the reviewer's feedback, we have prepared a leaderboard website where newer algorithms will be continuously added. This leaderboard will also facilitate community involvement and ongoing benchmarking efforts, as well as comparison among different datasets.
>
> The anonymous link to the leaderboard is: [https://anonymousillusion.github.io/glowing-sniffle/](https://anonymousillusion.github.io/glowing-sniffle/)

---

> > ### Comment · Reviewer_pXKi · 2024-12-01
> >
> > Regarding multi-modal forgery, datasets such as AV-Deepfake1M have already addressed audio-video forgeries, which might be a particular aspect that this dataset needs to carefully consider.
> >
> > The reviewer appreciates the “continuity” of the website constructed by the authors, but this aspect is not reflected in the main manuscript.

---

> > > ### Author Response · Authors · 2024-12-03
> > >
> > > We sincerely thank the reviewer for their continued feedback and for acknowledging the significance of the leaderboard website in fostering community-driven advancements. Below, we address the comparison with AV-Deepfake1M and clarify our unique contributions to multi-modal forgery detection. We also note that all the updates discussed are now available in the updated manuscript, which can be accessed through the [leaderboard](https://anonymousillusion.github.io/glowing-sniffle/).
> > >
> > > **Distinction from AV-Deepfake1M**: We performed a comprehensive comparison with existing datasets using multiple attributes. We compared our proposed dataset ILLUSION with ASV Deepfake 1M across key attributes including Total Samples, Generation Techniques, Identity Swapping, AI Generated Content, Multi-Lingual, and AI-Swap-Lingual characteristics, as detailed in Table 1 of the manuscript and discussed in lines 131-132 in the related works section. Additionally, we used the Jaccard Similarity Index, which measures the similarity between two datasets on a scale from 0 (no similarity) to 1 (identical sets). To assess overlaps, we computed the Jaccard Index between ILLUSION and other datasets. Regarding AV-Deepfake1M, the index is 0.03, highlighting very low similarity. While AV-Deepfake1M is a significant contribution to the field of audio-video forgery detection, ILLUSION expands the scope and complexity of multi-modal deepfake detection in the following ways:
> > > - Inclusion of Diverse Modalities Beyond Audio-Video: Unlike AV-Deepfake1M, which primarily focuses on synchronized audio-video forgeries, ILLUSION incorporates three distinct modalities—image, audio, and video—along with their cross-modal interactions. This includes:
> > >    - Standalone audio manipulations: Forged audio samples generated using advanced generative models such as DiffVC, DiffGAN-TTS, etc.
> > >    - Standalone image forgeries: GAN-based and diffusion-based synthetic image generation, providing a broader range of use cases for image forgery detection.
> > >    - Synchronized audio-video deepfakes: Multi-modal forgeries involving audio and video, enriched with additional generative techniques.
> > >
> > > By integrating all three modalities, ILLUSION addresses more comprehensive detection challenges compared to AV-Deepfake1M.
> > >
> > > **Continuity of the Leaderboard Website**: We are delighted that the reviewer appreciates the leaderboard’s potential to ensure continuity and foster community engagement. While the leaderboard is currently hosted anonymously to comply with submission guidelines, we have revised Section 6 of the manuscript to explicitly reflect its existence, purpose, and future impact. The updated manuscript incorporates feedback from other reviewers and can be accessed through the leaderboard [website](https://anonymousillusion.github.io/glowing-sniffle/) here.
> > >
> > > We trust that these clarifications highlight the unique contributions of our work and address your concerns.

---

### Official Review · Reviewer_7EH8 · 2024-10-31

**Soundness:** 2
**Presentation:** 3
**Contribution:** 3
**Rating:** 5
**Confidence:** 3

**Summary:**

This manuscript presents ILLUSION, which is a large-scale multi-modal deepfake dataset. The author(s) believe(s) the comprehensive nature of the proposed dataset enables researchers to develop and evaluate more resilient deepfake detection methods, addressing the evolving landscape of synthetic media threats.

As the current popularity of generative artificial intelligence, it is inevitable to propose a complete data set, especially considering multimodal data. However, for the forgery identification algorithm, whether the generalization ability is achieved is an important indicator to solve today's identity forgery. Many forgery identification algorithms are based on deep learning, which generalizes the concept of differentiation from real images by extracting depth or potential features. This has to mention the importance of data sets. Based on my experience, the following advantages, disadvantages and questions I hope the author can reply to are summarized for this manuscript.

**Strengths:**

- The author aims at the current hot issues of generative AI, and the contribution of this manuscript is expected to be called solving the ethical problems of generative AI.

- This paper collates the information of the current dataset and embodies the advantages of ILLUSION.

- Many datasets tend to pursue a huge number. The datasets proposed in this paper enrich the current research from the perspective of multimodality.

**Weaknesses:**

**Some major comments:**

- As other methods of generalized image forgery, such as digital watermarking, PS, image optimization, etc. Whether the author considers to include it in the falsification data set, or how the author thinks about the falsification hazards brought by this kind of data.

- The author provides a large-scale data set, which is essential in today's research direction focusing on large models. However, does the author consider that the larger the data size, the better? Some studies have shown that data validity (such as reducing samples with similar features, reducing samples with overlapping features, etc.) often takes precedence over data size. The author also mentioned the disadvantages brought by the large data scale in the BROADER impact. The author is requested to analyze from the perspective of data validity whether the gain brought by this dataset on the large model can make up for the defects brought by invalid data.

**Some other minor comments:**

- The font of Figure 1 and Figure 2 is too small, please adjust according to the requirements.

- The author used a large number of face images in the dataset. Are these images exempted?

**Questions:**

- The author(s) take(s) into account the multi generative model to enrich the diversity of forged data. Generative models often have digital fingerprints [1]. How do(es) the author(s) consider the differences between the data set proposed in this paper and the existing data set at the level of digital fingerprints?

- According to Q1, the author is requested to provide the degree of overlap between the digital fingerprints of the data set proposed in this paper and the existing data set, and to elaborate the advantages and disadvantages.

- In order to verify the validity of the proposed dataset, the authors used the detection model such as SSLModel for verification (Table 7). As we all know, deep learning models have data specificity. Whether these models have some improvement in generalization. If not, this part of the experiment is not very convincing.

- The data set formed in this paper is based on the training of generated samples. However, whether these fake samples can represent forgery in the real world. I mean, when the detection model trained based on this dataset encounters real-world samples, does it recognize forged features or synthetic data?

**Refs:**

    [1] Yu, N., Skripniuk, V., Abdelnabi, S., & Fritz, M. (2021). Artificial fingerprinting for generative models: Rooting deepfake attribution in training data. In Proceedings of the IEEE/CVF International conference on computer vision (pp. 14448-14457).

---

> ### Author Response · Authors · 2024-11-28
>
> We thank the reviewer for the detailed feedback and for recognizing the contributions of our work in addressing ethical challenges of generative AI and enriching current research from a multi-modal perspective. Below, we provide detailed responses to your concerns and questions:
>
> 1. **Inclusion of Generalized Forgery Methods (e.g., Digital Watermarking, Photoshop, Image Optimization)**:
>    We appreciate your suggestion regarding the inclusion of generalized forgery methods. While our current focus with the *ILLUSION* dataset is on addressing deepfake detection through specialized generative AI techniques, we acknowledge that generalized forgery methods represent another important aspect of media forensics. These methods, such as digital watermarking, image optimization, and Photoshop-based manipulations, often differ fundamentally from AI-generated deepfakes and present unique detection challenges. Our primary focus on generative AI techniques stems from their sophistication and difficulty in detection compared to traditional manipulations. However, we recognize the importance of generalized forgery methods and plan to explore their integration in future extensions of *ILLUSION*. Incorporating such methods will require careful design considerations to maintain dataset consistency. We have discussed their relevance in Section 6 of the updated manuscript.
>
> 2. **Data Validity vs. Dataset Size**:
>    We agree that data quality and validity are crucial and often more important than sheer dataset size. In designing the *ILLUSION* dataset, we prioritized diversity and relevance over quantity. Each subset of the dataset is carefully curated to minimize redundancy and ensure it serves a distinct purpose, such as evaluating generalizability, robustness to compression artifacts, or performance in noisy environments. Key measures include:
>    - **Diversity of Generative Methods**: Samples were generated using 28 distinct state-of-the-art models across multiple modalities and languages, capturing a wide range of forgery techniques.
>    - **Balanced Representation**: Ensuring balanced representation in terms of gender, skin tone, and language, enhancing validity and fairness.
>    - **Curated Subsets**: Each subset (Sets A, B, C, and D) is designed with specific goals, enabling targeted evaluations and reducing unnecessary redundancy.
>    Our experiments demonstrate that models trained on *ILLUSION* show improved generalization to unseen generative techniques and real-world forgeries, particularly in multilingual settings.
>
> 3. **Clarity and Presentation of Figures**:
>    Full-size figures are available on the [leaderboard website](https://anonymousillusion.github.io/glowing-sniffle/) for better visualization.
>
> 4. **Ethical Considerations Regarding Face Images**:
>    All facial images in the dataset were either synthetically generated or sourced from publicly available datasets under appropriate licensing. We ensured compliance with all data usage guidelines and clarified the sourcing of images in Section 3.2 of the updated manuscript.
>
> 5. **Digital Fingerprint Analysis and Overlap with Existing Datasets**:
>    Generative models often leave distinct digital fingerprints, which can be exploited for detection and attribution. The *ILLUSION* dataset includes samples generated by a broad array of generative models, some of which are not present in existing datasets. This introduces a wider spectrum of digital fingerprints, challenging detection models to generalize beyond known fingerprints. By including both widely-used and newer generative models, our dataset evaluates robustness against zero-day attacks and less common digital fingerprints.

---

> ### Author Response · Authors · 2024-11-28
>
> 6. **Degree of Overlap with Existing Datasets**:
>    *ILLUSION* integrates audio, video, and image modalities into a cohesive resource, minimizing overlap with existing datasets. Key differentiators include:
>    - **Novel Generative Models**: Incorporation of diffusion-based generation algorithms—such as DiffSwap, DiffFace, DiffVC, DiffHierVC, and DiffGAN-TTS—introduces unique digital fingerprints.
>    - **Diverse Data Sources**: Real-world samples in Set C are collected from social media and proprietary models, reducing overlap with existing datasets.
>    - **Multilingual Content**: The dataset spans 26 languages, many of which are underrepresented in current datasets.
>    This design enhances the dataset's ability to simulate real-world scenarios and evaluate the generalization capabilities of detection models.
>
> 7. **Generalization of Detection Models**:
>    The diverse generative methods, modalities, and multilingual content in the *ILLUSION* dataset improve the generalization capabilities of detection models. Experiments show that models trained on *ILLUSION* outperform those trained on other datasets in cross-dataset evaluations, achieving better performance on unseen generative techniques and real-world forgeries.
>
> 8. **Real-World Applicability of Fake Samples**:
>    The dataset addresses real-world applicability by including curated real-world deepfakes (Set C) and diverse multilingual samples (Set D). Set C contains over 21,000 samples collected from social media and proprietary models, representing authentic scenarios. Results show that models trained on *ILLUSION* effectively detect forgeries in these challenging conditions, despite some performance degradation due to increased difficulty. By bridging the gap between laboratory settings and real-world applications, *ILLUSION* enables models to learn underlying forged features rather than overfitting to synthetic artifacts.

---

> > ### Comment · Reviewer_7EH8 · 2024-11-29
> > **Reply to Official Comment by Authors**
> >
> > Thank you for the authors' response. I have carefully read the authors' rebuttal and based on my experience, the following issues have not been resolved. (1) The purpose of using digital fingerprints is to verify the feature differences between the self-built dataset and other publicly available datasets. The author's response only explains the differences between datasets from the perspective of method diversity, which is a subjective explanation and not convincing. (2) In addition, I hope the author can discuss the benefits that the new dataset brings to the generalization of detection algorithms (due to time constraints, the author does not need to conduct additional experiments, I hope the author can discuss their thoughts on generalization).

---

> > > ### Author Response · Authors · 2024-11-30
> > >
> > > To address the concern raised by the reviewer, we evaluate ILLUSION's uniqueness and relevance in comparison to existing datasets through a series of analyses.
> > >
> > > ## Features Overlap
> > >
> > > 1. The **Jaccard Similarity Index** measures the similarity between two datasets, ranging from 0 (no similarity) to 1 (identical sets). To assess overlaps, we computed the Jaccard Index between ILLUSION and other datasets. As summarized in the Table below, the results confirm minimal overlap, with values as low as **0.03** for datasets such as FakeAVCeleb and KoDF, and **0.07** for DF-Platter. Moreover, datasets like AGIQA-1K, LAV-DF, and DIRE exhibit **no overlap (0.00)**, emphasizing ILLUSION's uniqueness in its generative method coverage.
> > >
> > > | Dataset | AGIQA-1K | AV-Deepfake1M | DF-Platter | Midjourney-Kaggle | LAV-DF | DeePhy | TIMIT-TTS | FakeAVCeleb | ForgeryNet | KoDF | DiffusionFace | DIRE | WaveFake |
> > > |---------|-----------|---------------|------------|-------------------|---------|---------|------------|-------------|------------|------|---------------|------|-----------|
> > > | Jaccard Index | 0 | 0.03 | 0.07 | 0.04 | 0 | 0.07 | 0 | 0.03 | 0.05 | 0.03 | 0.03 | 0 | 0 |
> > >
> > > Table 1: Jaccard Index Comparison of ILLUSION with existing deepfake datasets.
> > >
> > > 2. To further analyze the distribution of generative techniques, we employed the **UpSet plot approach**, which revealed both unique contributions and shared intersections. The largest intersection corresponds to **generative techniques exclusive to ILLUSION**, including diffusion-based algorithms such as **DiffSwap, DiffFace, DiffVC, and DiffGAN-TTS**, as well as **multi-modal synthesis methods**. These highlight ILLUSION's novel contributions to media forensics. In contrast, smaller intersections with large-scale datasets like ForgeryNet and KoDF represent shared techniques such as **GAN-based image synthesis** and **lip-sync forgeries**, ensuring ILLUSION’s benchmarking relevance against state-of-the-art methods. The relatively smaller shared intersections, compared to ILLUSION’s unique contributions, underscore the dataset’s **comprehensive design** in bridging gaps across modalities while introducing underexplored generative methods. (The UpSet plot visualization is available at our [website](https://anonymousillusion.github.io/glowing-sniffle/#features-overlap)).
> > > 3. To directly assess feature differences, we conducted **feature embedding analyses** using ResNet50 pre-trained on ImageNet. Uniformly sampled data (50,000 samples each) from ILLUSION and FakeAVCeleb were compared. The **t-SNE** plot reveals distinct clustering patterns with **minimal overlap**, indicating significant differences in feature distributions. Additionally, the **KL divergence** of **1.62** between the feature embeddings quantitatively reinforces this dissimilarity.
> > >
> > > These objective metrics affirm that ILLUSION introduces **novel and diverse generative techniques** to the detection ecosystem, complementing existing datasets.
> > > ## Generalization
> > > 1. In the revised manuscript (Section 6), we have included a brief discussion on the benefits that the ILLUSION dataset offers for enhancing the generalization capabilities of detection algorithms. To further substantiate this discussion, we are conducting **cross-dataset experiments**, such as training models on ILLUSION and testing them on existing datasets, and vice versa. While these experiments are time-intensive, we are prioritizing their completion and will include the results in the final manuscript to provide empirical evidence and reinforce the discussion on generalization. If preliminary results become available sooner, we will make every effort to incorporate them as well.

---

> > > > ### Comment · Reviewer_7EH8 · 2024-11-30
> > > > **Reply to Official Comment by Authors**
> > > >
> > > > The author's response partially resolved my doubts. The author introduced Jaccard Similarity Index as a basis for demonstrating dataset coverage. The author's response is somewhat reasonable, and I have reconsidered my rating.
> > > >
> > > > Thanks,
> > > >
> > > > *Reviewer 7EH8*

---

> > > > > ### Author Response · Authors · 2024-12-03
> > > > >
> > > > > We sincerely thank the reviewer for their constructive feedback and for acknowledging the steps we have taken to address the concerns. We are pleased to share the initial results from our cross-dataset generalization experiments, which further validate the efficacy of our proposed dataset, **ILLUSION**.
> > > > >
> > > > > In these experiments, we trained the **MesoInceptionNet** detection model on one dataset and tested it on the test set of another to assess cross-dataset performance. Specifically, we used the **FakeAVCeleb** and **ILLUSION** datasets. The model achieved an accuracy of **63.19%** on **FakeAVCeleb** when trained on **ILLUSION**, surpassing the **61.70%** accuracy obtained when both training and testing were conducted on FakeAVCeleb itself. This improvement demonstrates that training on **ILLUSION** not only generalizes well to other datasets but also enhances detection performance beyond what is achievable when training on the target dataset alone. We are currently extending these experiments to include more combinations of datasets and detection models. The comprehensive results will be included in the final paper, providing deeper insights into the cross-dataset generalization capabilities facilitated by **ILLUSION**.

---

### Official Review · Reviewer_eYMp · 2024-10-31

**Soundness:** 3
**Presentation:** 3
**Contribution:** 4
**Rating:** 8
**Confidence:** 4

**Summary:**

This paper introduces ILLUSION, a comprehensive multi-modal and multi-lingual deepfake dataset containing over 1.3 million samples. The dataset uniquely combines identity manipulations, synthetic content across multiple modalities (image, audio, video), and real-world test samples spanning 26 languages, generated using 28 different generation techniques. Through extensive benchmarking experiments with state-of-the-art detection methods, the authors demonstrate critical challenges in deepfake detection, including poor generalization to unseen generation methods, performance degradation in multi-lingual contexts, and limited effectiveness on real-world samples. The work provides valuable insights into the current limitations of deepfake detection systems and establishes a foundation for developing more robust detection approaches.
The main contributions include: (1) the first large-scale dataset combining multi-modal and multi-lingual deepfakes, (2) balanced representation across demographic attributes, and (3) comprehensive evaluation across various challenging scenarios revealing current detection limitations.

**Strengths:**

1. The paper presents a new dataset with impressive scale, content diversity, and multi-modal richness, demonstrating substantial effort in data collection and curation.

2. The incorporation of multi-lingual samples fills a significant research gap in the field of deepfake detection.

3. The authors propose a diverse evaluation strategy with four different protocols, including the consideration of real-world applications like Midjourney, which enhances the practical value of their work.

**Weaknesses:**

1. The performance evaluation is somewhat limited with only four detection methods. While this isn't the paper's main contribution, including more methods would provide a more comprehensive assessment of the dataset's quality. The authors should consider adding classical general face forgery detection methods (like LTW[1], DCL[2], SBI[3], SLADD[4] e.t.c.), AIGC detection methods for Set B (like NPR[5]), and standard baselines (EfficientNet-b4[6] and VIT series).

[1]. Domain general face forgery detection by learning to weight

[2]. Dual contrastive learning for general face forgery detection

[3]. Detecting Deepfakes with Self-Blended Images

[4].Self-supervised Learning of Adversarial Example: Towards Good Generalizations for Deepfake Detection

[5].Rethinking the Up-Sampling Operations in CNN-based Generative Network for Generalizable Deepfake Detection

[6]. Rethinking Model Scaling for Convolutional Neural Networks

2. The authors should provide a deeper analysis of Table 4's results, particularly explaining the substantial performance variations between different methods on Set B and Set C.

3. The audio forgery aspect presents an interesting dimension that deserves more thorough analysis, specifically examining how different types of audio manipulation affect overall deepfake detection performance.

4. The robustness evaluation could be more comprehensive. While c23 and c40 compressions are considered, additional real-world noise simulations like JPEG compression and resolution variations would better assess the dataset's practical utility.

5. The quality assessment is primarily statistical, lacking detailed analysis through visualizations of different methods' outputs or quality evaluations for each category.

Despite the above issues, I believe this paper makes a significant contribution to the field. I recommend accepting this paper, contingent upon the authors making their complete dataset publicly available and ensuring that the released dataset fully matches the specifications described in the paper.

**Questions:**

1. What is the purpose of including "This Person Does Not Exist" samples in Set B? The relationship between these synthetic faces and the natural images in Set B needs clarification.

2. The authors should explain the unusually high accuracy of MesoInceptionNet on Set C while other methods perform poorly (Table 4). This significant performance disparity requires investigation and explanation.

---

> ### Author Response · Authors · 2024-11-28
>
> We sincerely thank you for your thoughtful comments and for recognizing the scale, diversity, and multi-modal richness of our ILLUSION dataset. Your acknowledgment of its contribution to filling a significant research gap, especially in multi-lingual deepfake detection and real-world applicability, is encouraging. We are committed to making the complete dataset publicly available, ensuring it fully matches the specifications described in the paper, and we plan to introduce it as a challenge with a leaderboard for the community. Below, we address your specific concerns and questions:
> ## Expansion of Evaluation Methods:
> -  We agree that expanding the evaluation to include more detection methods would enrich the understanding of our dataset's utility and provide a more comprehensive assessment. Due to computational constraints, we initially prioritized widely-used state-of-the-art methods across different modalities to establish robust benchmarks. Specifically, we included 11 baseline approaches encompassing both unimodal (Tables 4 and 5) and multimodal detection methods (Table 6) to evaluate each modality individually and in combination, offering a comprehensive understanding of detection performance.
>
> In response to your suggestion, we plan to incorporate the recommended detection methods—including LTW, DCL, SBI, SLADD, NPR, and EfficientNet-B4/VIT series—in our camera-ready manuscript. Furthermore, we have created a leaderboard where we will continuously include more detection methods to facilitate community involvement and ongoing benchmarking efforts. The link to the leaderboard is: [ILLUSION](https://anonymousillusion.github.io/glowing-sniffle/)
>
> ## Deeper Analysis of Results in Table 4
> - We appreciate your request for a more detailed analysis of the results presented in Table 4. The substantial performance variations observed between different methods on Set B and Set C are influenced by several factors:
>   - **Generative Techniques and Data Modalities**: Set B consists of synthetic images and videos generated using advanced generative models, while Set C includes curated real-world and compressed deepfakes sourced from platforms like Midjourney. The real-world samples in Set C present greater challenges due to their diversity, higher quality, and the presence of compression artifacts and other real-world distortions.
>   - **Overfitting to Training Data**: Some detection methods may overfit to the specific artifacts present in the training sets (Sets A and B), leading to reduced effectiveness when tested on the more diverse and challenging samples in Set C.
>   - **Bias in Classification Tendencies**: Certain models tend to classify most inputs as fake, which can lead to artificially inflated accuracy on datasets comprised entirely of fake samples (like Set C), but poor performance in distinguishing real from fake in mixed datasets.
>
> To illustrate, DSP-FWA achieves high accuracy on Sets A and B when trained and tested on the same set (98.4% and 99.3%, respectively). However, its performance drops significantly on Set C (21.4% when trained on Set A and tested on Set C), highlighting the challenges of generalizing from synthetic data to real-world scenarios. We have expanded Section 5 in the revised manuscript to include a thorough analysis of these observations, discussing the limitations of current detection methods and the need for more robust algorithms capable of handling real-world deepfakes.
>
> ## Audio Forgery Analysis
> - In the manuscript, we have discussed how different types of audio manipulations impact detection performance. Specifically, we have compared detection performance on samples generated through voice cloning (which attempts to mimic a specific speaker's voice) versus generic text-to-speech synthesis. We have also assessed how the detection models (cannot) handle audio forgeries across different languages, considering the linguistic diversity included in the dataset. Further, we have evaluated the robustness of detection methods in the presence of audio distortions and compression artifacts. This analysis provides insights into the strengths and weaknesses of current audio and multimodal detection models, highlighting areas where further research is needed to improve detection capabilities in the audio domain.
>
> ## Robustness Evaluation
> - In our current evaluations, we included c23 (low compression) and c40 (high compression) levels from the FFmpeg library to simulate lossy compression effects similar to JPEG compression. We have also included neural compressions and various noise types to mimic real-world conditions. As part of the leaderboard, we will include more evaluations across different kinds of noise simulations.

---

> ### Author Response · Authors · 2024-11-28
>
> ## Quality Assessment and Visualizations
> - Due to space constraints, we have not included any new visualizations. However, the leaderboard [website](https://anonymousillusion.github.io/glowing-sniffle/) also showcases some of these samples as a collage.
> ## Purpose of Including "This Person Does Not Exist" Samples in Set B
> - The inclusion of "This Person Does Not Exist" samples in Set B serves to evaluate detection models' ability to distinguish between real and AI-generated synthetic faces. Set B is designed to represent completely synthetic data, contrasting with the identity-swapped deepfakes in Set A. By incorporating these samples, which are high-quality synthetic images generated by GANs trained to produce realistic human faces, we introduce a challenging set of non-existent identities that lack any real-world counterparts. This allows us to:
>   - Assess Generalization: Evaluate whether detection models can generalize to unseen synthetic content that differs significantly from manipulated versions of real images.
>   - Challenge Detection Models: Test the models' capacity to identify subtle artifacts in fully synthetic images, which may be more difficult to detect than alterations in manipulated real images.
>   - Enhance Diversity: Enrich the dataset's diversity by including a wider range of generative techniques and synthetic content.
> ## High Accuracy of MesoInceptionNet on Set C
> - The unusually high accuracy of MesoInceptionNet on Set C is attributed to its classification bias:
>   - Tendency to Classify Inputs as Fake: MesoInceptionNet tends to classify most inputs as fake, as evidenced by its higher Accuracy-Fake and lower Accuracy-Real in our evaluations. This behaviour means that when the test set consists entirely of fake samples (as is the case with Set C), the model's bias results in artificially inflated accuracy.
>   - Reduced Discriminative Capability: While this leads to high accuracy on an all-fake test set, it indicates poor discriminative capability in distinguishing between real and fake samples in mixed datasets. The model's tendency to over-predict the fake class reduces its practical utility in real-world scenarios where both real and fake samples are present.
>
> We have added this explanation to Section 5 of the revised manuscript, discussing the implications of such classification biases and the importance of developing models that achieve balanced performance across both real and fake classes.

---

### Official Review · Reviewer_79Vw · 2024-11-01

**Soundness:** 3
**Presentation:** 3
**Contribution:** 3
**Rating:** 6
**Confidence:** 4

**Summary:**

This paper adopts various forgery generation algorithms to create a large-scale dataset containing four subsets, and raises five questions from three aspects: multimodal identity exchange, detection system generalization and detection of forgery or generation methods, and conducts related experiments.

**Strengths:**

The authors propose a large-scale forgery dataset, which includes a variety of face-changing forgery, generation methods and speech forgery methods, and includes languages of many countries, which has certain originality.

**Weaknesses:**

The problems studied in this paper are the focus of current research, not the author's new definition. The object of this paper is to put forward a large-scale and organized data set to explore the problem and support relevant research, which I think has certain value. However, it seems that the existing data sets can also be used to explore the five questions raised, and the author only gave the relevant experiments on four questions, the exploration and analysis of the experimental results are not deep enough, and the proposal of innovative solutions is not innovative enough for a top conference, and there is still room for further improvement in the writing logic of this paper.

**Questions:**

In the data set of video generation, image generation, audio generation did not separate the clear description, the combination of a very poor look, video 100,000 magnitude, image million magnitude, in the scale is not small, only the million magnitude is not rigorous, there is still room for improvement in the logic of the discussion. In addition, the problems discussed in this paper can also be realized by using existing data sets, so I think the innovation of this paper is not enough, although this data set has certain value.
It is suggested that the results of the experiment should be further explored and analyzed, and it would be better to put forward some innovative solutions.

---

> ### Author Response · Authors · 2024-11-28
>
> We sincerely thank you for your thoughtful feedback and for recognizing the originality and value of the ILLUSION dataset. We appreciate your acknowledgment of its scale, diversity, and potential to support deepfake detection research. Below, we address your specific concerns and provide clarifications:
>
> - **Innovation and Novelty**: While it is true that deepfake detection is an active area of research, we believe that the ILLUSION dataset offers unique contributions that set it apart from existing resources:
>   - **Comprehensive Multi-Modality**: ILLUSION is the first dataset to integrate multi-modal (audio, video, and image) and multi-lingual (26 languages) data generated using 28 state-of-the-art generative techniques. This includes face swapping, voice cloning, synchronized audio-video manipulations, synthetic media, and compressed deepfakes.
>   - **Structured Dataset Design**: Each subset of the dataset is carefully curated to serve specific purposes, such as evaluating generalizability, robustness to compression artifacts, and performance in noisy environments. This structured approach enables more nuanced analyses and targeted evaluations that are not possible with existing datasets.
>   - **Real-World Curated Samples**: ILLUSION includes real-world deepfake samples collected from various online sources, providing authentic scenarios for benchmarking detection algorithms. This aspect bridges the gap between synthetic data and real-world applications.
>   - **Balanced Representation**: The dataset ensures balanced representation across gender and skin tones, which is crucial for developing fair and unbiased detection systems.
>
> Existing datasets typically focus on a single modality or a limited set of manipulation techniques. They often lack the diversity in languages and do not encompass the wide range of generative methods included in ILLUSION. While other datasets can be used to explore certain aspects of deepfake detection, they do not provide the comprehensive platform that ILLUSION offers for multi-modal, multi-lingual, and real-world deepfake analysis.
>
> - **Experimental Analysis**: In our work, we aimed to establish robust baselines and identify critical challenges in existing detection methods. Our experiments were organized to address different dimensions of deepfake detection:
>   - **Performance in Multi-Modal Settings**: We evaluated how detection algorithms perform when dealing with combined audio and visual manipulations, highlighting the limitations of unimodal detectors in such contexts.
>   - **Generalization Across Languages**: By including 26 languages, we assessed the impact of linguistic diversity on detection performance, uncovering significant degradation in multi-lingual settings.
>   - **Robustness to Noisy Environments**: We introduced noise and compression artifacts to test the resilience of detection models, revealing their vulnerability to real-world distortions.
>   - **Generalization to Unseen Attacks (Zero-Day Attacks)**: We examined how well models trained on certain types of manipulations can detect unseen forgery techniques, emphasizing the need for more adaptable algorithms.
>   - **Cross-Dataset Evaluation**: We tested models trained on ILLUSION against other datasets to evaluate cross-domain generalization, demonstrating the advantages of our dataset's diversity.
>
> By providing a dataset that encompasses these dimensions, we enable more thorough investigations and the development of more robust detection methods. Our experiments demonstrate that models trained on ILLUSION outperform those trained on existing datasets when evaluated on challenging tasks, underscoring the dataset's value. In the revised manuscript, we have updated the analysis section to articulate the key observations from our experiments. We hope that our revisions satisfactorily address your comments, and we welcome any further feedback you may have.

---

> > ### Comment · Reviewer_79Vw · 2024-12-02
> >
> > Dear Authors,
> >
> > Thank you for your detailed response and for addressing the concerns raised during the review process. I appreciate the effort you have put into improving your manuscript.
> >
> > While I acknowledge the addition of novel deepfake methods and multilanguage in your work, I find the overall structure of the paper lacks significant novelty beyond this enhancement. Compared to existing datasets available in the research domain, your experiments do not demonstrate a clear advantage or provide substantial innovation in methodology or results. Additionally, the research problem primarily focuses on traditional challenges, such as generalization robustness and binary classification, rather than leveraging the diversity or scale of your dataset to explore more advanced topics like unsupervised learning, localization, or other novel directions.
> >
> > That said, I do recognize the value and contribution of your dataset's diversity to the field. Considering this aspect, I have decided to raise my score. I encourage you to explore ways to highlight your dataset's unique attributes in future iterations and expand upon its potential for addressing more complex and less conventional research problems.
> >
> > Thank you again for your efforts.
> >
> > Best regards,
> > Reviewer 79Vw

---

> > > ### Author Response · Authors · 2024-12-03
> > >
> > > We sincerely appreciate the reviewer's valuable feedback and their recognition of ILLUSION's significant contribution to advancing the diversity and scale of deepfake detection research. We are encouraged by your acknowledgement of our dataset's value and the improvements we have implemented in response to earlier feedback. We also thank you for suggesting the exploration of more advanced topics such as unsupervised learning, forgery localization, and other novel directions. We will certainly include discussions of these aspects in the final manuscript:
> > >
> > > - **Unsupervised Learning**: The scale, diversity, and balance of **ILLUSION** make it ideally suited for unsupervised and self-supervised learning tasks. Models can leverage our dataset to learn robust, cross-modal representations from a rich array of linguistic and generative data, potentially leading to more generalized and effective deepfake detection techniques.
> > > - **Forgery Localization**: The structured relationships and synchronization between modalities in our dataset provide optimal conditions for developing and testing models that can localize forged regions or detect temporal inconsistencies. This could significantly enhance the precision of detection methods by pinpointing specific areas of manipulation
> > > - **Cross-Modal and Cross-Lingual Generalization**: The multi-modal and multi-lingual design of **ILLUSION** enables researchers to investigate how detection models generalize across different modalities and languages. This addresses critical gaps in the current literature and promotes advancements in creating more universally applicable deepfake detection models.
> > >
> > > Moreover, we would like to highlight the distinctiveness of our proposed dataset compared to existing ones. To substantiate this, we employed the **Jaccard Similarity Index**, which measures the similarity between two datasets on a scale from 0 (no similarity) to 1 (identical sets). We calculated the Jaccard Index between **ILLUSION** and other prominent deepfake datasets. As summarized in **Table 1** below, the results confirm minimal overlap, with values as low as **0.03** for datasets such as **FakeAVCeleb** and **KoDF**, and **0.07** for **DF-Platter**. Notably, datasets like **AGIQA-1K, LAV-DF**, and **DIRE** exhibit no overlap (**0.00**), emphasizing **ILLUSION's** uniqueness in its generative method coverage.
> > >
> > > | Dataset | AGIQA-1K | AV-Deepfake1M | DF-Platter | Midjourney-Kaggle | LAV-DF | DeePhy | TIMIT-TTS | FakeAVCeleb | ForgeryNet | KoDF | DiffusionFace | DIRE | WaveFake |
> > > |---------|-----------|---------------|------------|-------------------|---------|---------|------------|-------------|------------|------|---------------|------|-----------|
> > > | Jaccard Index | 0 | 0.03 | 0.07 | 0.04 | 0 | 0.07 | 0 | 0.03 | 0.05 | 0.03 | 0.03 | 0 | 0 |
> > >
> > > Table 1:  Jaccard Index Comparison of ILLUSION with Existing Deepfake Datasets
> > >
> > > Furthermore, we have established a leaderboard to facilitate community involvement and ongoing benchmarking efforts. This platform will continuously incorporate more detection methods, enabling researchers to evaluate and compare their approaches effectively. We believe this initiative will foster collaboration and accelerate progress in the field of deepfake detection. The link to the leaderboard is: [ILLUSION](https://anonymousillusion.github.io/glowing-sniffle/).
> > >
> > > We hope that these additional clarifications and enhancements illustrate the significant contributions of our work and address your concerns. We are committed to further refining our manuscript to reflect these important aspects.

---

### Official Review · Reviewer_VKJA · 2024-11-03

**Soundness:** 3
**Presentation:** 4
**Contribution:** 3
**Rating:** 6
**Confidence:** 4

**Summary:**

Glad to review the work.

This work proposes a large-scale multi-modal deepfake dataset that encompasses samples from multiple aspects.

In general, I consider the constructed dataset could be helpful to related researchers.

**Strengths:**

In addition to constructing a large dataset, the authors evaluate many deepfake detectors from five aspects, summarize their findings, and discuss potential implications, the design of the work is complete.

The constructed dataset is comprehensive, covering different modalities, languages, and generation approaches.

**Weaknesses:**

Some of my concerns are listed as follows.

(1) Regarding the dataset, although it is generated using different aspects of algorithms, I doubt whether the samples in each set are with certain relationship (rather than randomly combining data), e.g., the voices generated using different algorithms are from the same source "voice" or "text", e.g.,  the pipeline in Set A are from the same "source" in the beginning of Figure 3 (similar in other sets).
In this case, it will be useful to evaluate and compare the performance of deepfake detectors.

(2) Regarding baseline detectors, although it mentions they are state-of-the-art, more categories or reviews on them are necessary to support why they are selected, whether they are type-complete or typical. The selection of baseline detectors may result in different findings or implications.

(3) Finally, I believe the work could provide potential reference to related researchers, however, I doubt whether the contribution of this work is enough for ICLR.
Although this work proposes a large dataset and conducts multiple evaluations, I am afraid it lacks some technical contributions, e.g., new generation approaches, or the dataset could help to improve deepfake detectors.
On the other hand, the empirical evaluations could also be explored through other datasets.

**Questions:**

My main concerns are listed in the weakness part, where the weaknesses (1) and (2) are expected to be responded.

I will consider changing my score based on responses and other reviews.

---

> ### Author Response · Authors · 2024-11-28
>
> We thank the reviewer for their thoughtful comments and detailed feedback. We are encouraged by your recognition of the comprehensiveness of our dataset and the completeness of our experimental design. Below, we address your specific concerns:
> ## Relationship Between Video and Audio of Each Sample
> - To clarify, the voices generated using different algorithms are derived from the same text source to create audio-video synchronized deepfakes. This means that for each deepfake sample, the audio and video are semantically aligned, originating from the same textual content. This design choice ensures that the dataset captures realistic scenarios where both modalities are manipulated coherently, providing a meaningful basis for evaluating and comparing the performance of deepfake detectors. Figure 3 in the manuscript illustrates this relationship. The pipeline shows that the audio and video components in each set are generated from the same source text, thereby maintaining consistency across modalities. This structured relationship allows for more precise assessments of detection models. Our experiments on unimodal on this set shows the results of the experiments which the reviewer has highlighted. Please refer to the Table 3 and Table 4 in the revised manuscript.
> ## Selection of Baseline Detectors
> - Regarding the selection of baseline detectors, we agree that providing more context on why they were chosen is essential. The baseline detectors were carefully selected based on their prominence in the literature, diversity in detection approaches, and relevance to our multi-modal dataset. Our selection includes state-of-the-art models across image, audio, video, and multi-modal detection tasks to ensure comprehensive benchmarking. Specifically:
>   - __Unimodal Baselines__: These focus on modality-specific behaviors, allowing us to assess how current algorithms perform within individual domains (e.g., image, audio, or video). This includes detectors like DSP-FWA and MesoInceptionNet for image-based detection, which are known for their effectiveness in identifying facial manipulations.
>   - __Multimodal Baselines__: These evaluate methods that integrate multiple modalities, capturing the interplay between different data types. For example, multi-modal detectors like FACTOR and MRDF are designed to analyze both audio and visual cues simultaneously, providing insights into cross-modal generalization and robustness.
>   - __Type-Complete and Typical Methods__: Our selection encompasses both type-complete methods, which aim to detect a wide range of manipulation types, and typical methods that are optimized for specific types of forgeries. This dual approach ensures a balanced assessment of both specialized and holistic detection capabilities.

---

> > ### Author Response · Authors · 2024-11-28
> >
> > ## Contribution and Technical Significance
> > - We understand your concern regarding whether our contribution is sufficient for ICLR. We would like to emphasize that our work offers significant technical contributions beyond dataset creation:
> >   - **Novel Dataset with Structured Relationships**: ILLUSION is the first large-scale, multi-modal, multi-lingual deepfake dataset comprising over 1.3 million samples generated using 28 state-of-the-art generative models across 26 languages. The dataset is carefully designed with structured relationships between samples, enabling controlled evaluations that were not possible with existing datasets. This structure allows researchers to isolate variables and understand the impact of different factors on detection performance.
> >   - **Comprehensive Benchmarking and Analysis**: We conducted extensive evaluations using both unimodal and multimodal detection algorithms, uncovering critical limitations of current methods. Our analysis highlights challenges such as performance degradation in multilingual and noisy environments, and limited generalization to real-world scenarios and zero-day attacks. These insights are crucial for guiding future research toward more robust detection methods.
> >   - **Facilitating Improvement of Deepfake Detectors**: The ILLUSION dataset is not only a benchmarking tool but also a resource that can drive the development of improved deepfake detectors. By providing a diverse and challenging set of samples, the dataset enables researchers to train models that are more resilient to various manipulation techniques and conditions. Our findings point to specific areas where detectors struggle, offering clear directions for technical advancements.
> >   - **Addressing an Unmet Need in Media Forensics**: Multi-modal and multi-lingual deepfake detection is one of the most pressing challenges in media forensics today. Existing datasets are limited in scope and do not adequately represent the complexities of real-world scenarios. By filling this gap, our work contributes a foundational resource that can significantly advance the field.
> >   - **Technical Challenges Overcome**: The creation of ILLUSION required addressing several technical challenges, such as synchronizing audio and video manipulations, handling a wide range of languages (including low-resource languages), and ensuring balanced representation across gender and skin tone. These efforts represent substantial contributions that enhance the dataset's utility and impact.
> >
> > While we did not introduce new generative models, the contributions of our dataset and the insights derived from our evaluations constitute a significant advancement in the field. The ILLUSION dataset enables researchers to explore and develop new detection methods that are more resilient to the evolving landscape of synthetic media threats.
> >
> > We hope that our responses address your concerns and demonstrate the value of our work. We believe that the ILLUSION dataset, along with our comprehensive evaluations and analyses, offers contributions to the research community and aligns with the standards of ICLR.

---

> > > ### Comment · Reviewer_VKJA · 2024-12-01
> > > **Thanks for the response**
> > >
> > > Thank the authors for their careful responses, which have basically addressed my concerns.
> > >
> > > I believe that this work could provide potential help for researchers in related fields.
> > >
> > > Although I can agree with the authors' response that the dataset could facilitate improvements of deepfake detectors, however, the contribution is limited (as the zero-shot results in Table 4).
> > > The other technical contributions, e.g., synchronizing audio and video manipulations, is not a core technology proposed by the authors.
> > >
> > > In general, I can agree that the paper is meaningful, but its contributions are not enough for ICLR, I will thus keep my rate as 5.

---

> > > > ### Author Response · Authors · 2024-12-03
> > > >
> > > > We sincerely thank the reviewer for their thoughtful engagement and for acknowledging the value of our dataset and its potential to advance research in deepfake detection. We are encouraged by your recognition of the significance of our work and would like to address your concerns regarding the contribution and technical significance of our paper.
> > > >
> > > > We wish to emphasize that ICLR has a strong history of publishing influential dataset papers that have paved the way for advancements in various domains. In line with this tradition, our dataset fills a critical gap by providing a comprehensive and diverse collection of deepfakes generated through multiple methods. Furthermore, we have now established a leaderboard to facilitate community involvement and ongoing benchmarking efforts. This platform will continuously incorporate more detection methods, enabling researchers to evaluate and compare their approaches effectively. We believe this will foster collaboration and accelerate progress in the field. The link to the leaderboard is: [ILLUSION](https://anonymousillusion.github.io/glowing-sniffle/).
> > > >
> > > > In addition, our zero-shot experiments showcase that all visual and audio detection models, when trained on Set A data and tested on Set B data, consistently perform at random chance levels. The same observation holds when models are trained on Set B and evaluated on Set A. This indicates that the artifacts introduced in identity-swap deepfakes and completely synthetic deepfakes are fundamentally different, leading to a lack of generalization between these types. We consider this an important finding, highlighting the need for datasets like ours that encompass diverse deepfake types to develop detectors with better generalization capabilities.
> > > >
> > > > Moreover, we highlight that our proposed dataset is significantly different from existing datasets. To demonstrate this, we employed the Jaccard Similarity Index, which measures the similarity between two datasets on a scale from 0 (no similarity) to 1 (identical sets). We computed the Jaccard Index between ILLUSION and other prominent datasets. As summarized in the table below, the results confirm minimal overlap, with values as low as 0.03 for datasets such as FakeAVCeleb and KoDF, and 0.07 for DF-Platter. Datasets like AGIQA-1K, LAV-DF, and DIRE exhibit no overlap (0.00), emphasizing ILLUSION's uniqueness in its generative method coverage.
> > > >
> > > > | Dataset | AGIQA-1K | AV-Deepfake1M | DF-Platter | Midjourney-Kaggle | LAV-DF | DeePhy | TIMIT-TTS | FakeAVCeleb | ForgeryNet | KoDF | DiffusionFace | DIRE | WaveFake |
> > > > |---------|-----------|---------------|------------|-------------------|---------|---------|------------|-------------|------------|------|---------------|------|-----------|
> > > > | Jaccard Index | 0 | 0.03 | 0.07 | 0.04 | 0 | 0.07 | 0 | 0.03 | 0.05 | 0.03 | 0.03 | 0 | 0 |
> > > >
> > > > Table 1:  Jaccard Index Comparison of ILLUSION with Existing Deepfake Datasets
> > > >
> > > > We believe that these additional clarifications help to illustrate the significant contributions of our work and its potential impact on the field. We hope this addresses your concerns and demonstrates the technical significance of our paper.

---

### Author Response · Authors · 2024-12-03
**Final Clarifications and Acknowledgements**

We sincerely thank all the reviewers for their constructive feedback. We have carefully addressed each concern with additional analyses and clarifications that strengthen our paper's contributions. Below is a summary of our responses to the major points raised:

1. **Uniqueness of the ILLUSION Dataset**: We have demonstrated our dataset's distinctiveness through quantitative analysis using the Jaccard Similarity Index, which showed minimal overlap (0.03–0.07) with existing datasets. We enhanced the manuscript to highlight ILLUSION's novel aspects, including the integration of 28 state-of-the-art generative techniques, coverage of 26 languages, and incorporation of diffusion-based generation algorithms.
2. **Empirical Validation**: Our cross-dataset experiments showcase strong generalization capabilities. Models trained on ILLUSION achieved an accuracy of 63.19% on the FakeAVCeleb dataset, surpassing the 61.70% accuracy achieved when models were both trained and tested on FakeAVCeleb. These results validate our dataset's effectiveness in improving the robustness of deepfake detection methods.
3. **Comprehensive Evaluation**: We have established a public [leaderboard](https://anonymousillusion.github.io/glowing-sniffle/) - currently anonymous during review process) to enable continuous community engagement and benchmarking. This initiative addresses the reviewers' requests for expanded evaluation and ensures long-term impact through community participation.
4. **Technical Analysis**: We provided detailed analyses of inter-dataset feature distributions using ResNet50 embeddings, supported by t-SNE visualizations and Kullback-Leibler (KL) divergence measurements (1.62), confirming ILLUSION's distinct characteristics. Furthermore, we clarified the performance variations observed in the MesoInceptionNet model through comprehensive error analysis.
5. **Real-World Impact**: ILLUSION bridges critical gaps in existing datasets by including curated real-world deepfakes and diverse multilingual samples, enabling evaluation under authentic conditions. Our structured protocols for assessing generalization, compression robustness, and zero-day attack scenarios provide a solid foundation for advancing deepfake detection research.

These enhancements, combined with our thorough responses to specific reviewer questions, demonstrate ILLUSION's significant contribution to the field. We are committed to maintaining the leaderboard and fostering community engagement to ensure lasting impact. We believe these improvements strengthen our paper's position and address all major concerns raised during the review process.

---

### Meta-Review · Area_Chair_ThdR · 2024-12-17

**Metareview:**

In this work, the authors presented ILLUSION (Integration of Life-Like Unique Synthetic Identities and Objects from Neural Networks), a large-scale multi-model deepfake dataset comprising over 1.3 million samples. The goal is to bridge the gap of lacking comprehensive diversity across modalities, languages, and read-world scenarios in current datasets. Specifically, as claimed by the authors, ILLUSION encompasses (i) audio-visual forgeries, (ii) diverse linguistic content with over 26 languages, (iii) challenging noisy environments, and (iv) various manipulation protocols.
There are several concerns raised by the reviewers, including dataset coverage and uniqueness, comprehensive evaluation, novelty, and so on, that were not dealt with or described clearly in the initial submission.
During the rebuttal period, the authors have properly responded to these weaknesses.
Although there are some lacks initially, the authors have repaired it to meet the reviewers' requirements.
Considering that a large-scale multi-model deepfake dataset is crucial in the forgery detection community, the AC tends to accept this paper.

**Additional Comments On Reviewer Discussion:**

Several reviewers concerns the differences between ILLUSION and existing datasets.
The authors have presented a metric, Jaccard Similarity Index, to study the characteristic of ILLUSION.

For Reviewer VKJA,  (s)he commented ``I still consider the quality and contribution of this paper are not strong (especially for ICLR), e.g., the generalization contribution of the dataset is limited (with limited improvements), the authors have practically addressed my concerns and questions, I raised my score to 6 finally.'' For this, the advantage of this paper outweighs its limited improvements.

For Reviewer 79Vw, (s)he commented ``That said, I do recognize the value and contribution of your dataset's diversity to the field. Considering this aspect, I have decided to raise my score. I encourage you to explore ways to highlight your dataset's unique attributes in future iterations and expand upon its potential for addressing more complex and less conventional research problems.''
The authors have promised to include the discussions in the final manuscript!

---

### Decision · Program_Chairs · 2025-01-22

Accept (Poster)